# Exponential Map Models as an Interpretable Framework for Generating Neural Spatial Representations

## Abstract

A fundamental challenge in neuroscience and AI is understanding how physical space is mapped into neural representations. While artificial neural networks can generate brain-like spatial representations, such as place and grid cells, their "black-box" nature makes it difficult to determine if these representations arise as general solutions or as artifacts of a chosen architecture, objective function, or training protocol. Critically, these models offer no guarantee that learned solutions for core navigational tasks, like path integration (updating position from self-motion), will generalize beyond their training data. To address these challenges, we introduce a first-principles framework based on an exponential map model. Instead of using deep networks or gradient-based optimization, the presented model uses generator matrices to map physical locations into neural representations through the matrix exponential, creating a transparent framework that allows us to identify several exact algebraic conditions underlying key properties of neural maps. We show that path invariance (ensuring location representations are independent of traversal route) and *exact* path integration is achieved if the generators commute, while translational invariance (maintaining consistent spatial relationships across locations) demand that generators produce orthogonal transformations. We also show that preserving the metric of flat space requires the eigenvalues of the generator matrices to form sets of roots of unity. Finally, we demonstrate that the proposed framework constructs diverse biologically relevant spatial tuning, including place cells, grid cells, and context-dependent remapping, and that the exponential map model corresponds to the on-manifold dynamics of a continuous attractor network. The framework we propose thus offers a transparent, theoretically-grounded alternative to "black-box" models, revealing the exact conditions required for a coherent neural map of space.

## 1 Introduction

A fundamental challenge in neuroscience and artificial intelligence is to understand the mapping from physical space to the representational space of neural population activity. In the mammalian brain, such representations are strongly associated with the hippocampal formation, which contains specialized neurons that encode spatial information. Most famously, place cells (O'Keefe & Dostrovsky, 1971) fire within specific, localized areas of an environment known as place fields, while grid cells (Hafting et al., 2005) fire in a periodic hexagonal pattern that tessellates the environment and is hypothesized to provide a neural metric for space (McNaughton et al., 2006; Moser & Moser, 2008; Ginosar et al., 2023). Together, these and other spatially-tuned cells form a high-dimensional representation of an animal's location. This neuronal spatial map abruptly reorganizes in response to environmental changes, a phenomenon known as remapping, indicating that neurons also encode the environment's identity (Leutgeb et al., 2004; Fyhn et al., 2007). While the firing patterns of these spatial neurons are well-characterized, the principles governing their emergence remain unclear.

In recent years, deep learning models, particularly recurrent neural networks (RNNs), trained to solve navigation tasks have been shown to learn representations that resemble biological place and grid cells (Banino et al., 2018; Cueva & Wei, 2018; Sorscher et al., 2023). Complementing these end-

to-end approaches, more structured architectures have demonstrated that such representations can arise from explicit theoretical constraints, such as the factorization of abstract structural knowledge from specific sensory experiences (Whittington et al., 2020). In parallel, theoretical frameworks rooted in reinforcement learning, most notably the Successor Representation (SR), have proposed that grid cells function as a low-dimensional eigenbasis for the environment's transition matrix, effectively encoding a predictive map of space (Stachenfeld et al., 2017). Although these findings strongly suggest that spatial tuning is a normative solution to the demands of navigation, critical theoretical limitations remain. The "black-box" nature of deep neural networks makes it difficult to disentangle whether their learned representations reflect fundamental principles of navigation or are artifacts of a chosen architecture, objective function, or training protocol (see Fig. 1a) for an illustration). Furthermore, while SR and structured models improve interpretability, they share a fundamental limitation with deep learning approaches: they all rely on iterative optimization or statistical accumulation. Consequently, these methodologies do not offer an algebraic guarantee that navigational solutions will strictly generalize beyond the training data. In contrast, animals are able to seamlessly navigate vast, novel environments. To understand how biological brains solve these navigational challenges efficiently and robustly, there is a need for models that allow for exact and interpretable solutions to navigation problems.

In this work, we construct spatial representations using a first-principles framework based on the matrix exponential. Instead of relying on neural networks or gradient-based optimization, the presented approach builds representations from a transparent mathematical foundation. The core component of the model is a set of generator matrices that directly map a spatial location to a neural population firing rate vector. This construction enables us to derive the exact algebraic conditions required for a coherent neural spatial map. For a neural spatial map to be useful, it must support core navigational computations. One of the most fundamental is *path integration*, the process by which a navigator estimates its position by integrating self-motion cues. This process introduces a critical self-consistency problem: For the map to be coherent, the representation of a location must be independent of the path taken to reach it. We show that path-independent representations required for reliable path integration are guaranteed if the model's generator matrices commute. Furthermore, we find that equinorm representations, previously used as a learning constraint in neural networks (Schaeffer et al., 2023; Xu et al., 2022), arise naturally from generators that produce translationally invariant similarity structures—a desirable property for navigation in open-field environments. We also show that preserving the metric of flat space (Xu et al., 2022; Schøyen et al., 2025; Xu et al., 2025) requires the eigenvalues of the generator matrices to form sets of roots of unity on discrete rings in frequency space. Crucially, when these conditions are met, the framework recovers diverse, biologically observed tuning curves, such as place cells and grid cells, dictated strictly by the symmetry of the generators. We also demonstrate that this framework can be seamlessly generalized from preserving the metric of space to preserving the similarity of more general inputs, which we use to model remapping, and even goal-oriented, multi-map navigation in an interpretable manner. A conceptual overview of the proposed framework and the key spatial map properties we address are presented in Fig. 1. Finally, we demonstrate that the exponential map can be interpreted as describing the on-manifold dynamics of a continuous attractor recurrent neural network.

Despite its simplicity, the proposed framework is powerful enough to construct a variety of biologically plausible tuning curves, including place cells, grid cells, and context-dependent remapping, from the same underlying mechanism. By grounding spatial representations in a clear algebraic structure, the presented work provides a theoretically-grounded alternative to black-box models, revealing exact and interpretable principles that underpin a coherent neural map of space.

## 2 RESULTS & DISCUSSION

### 2.1 CONSTRUCTING SPATIAL REPRESENTATIONS WITH AN EXPONENTIAL MAP

Formally, a spatial representation is a map that assigns a neural population vector to every spatial location, as exemplified in Fig. 1b). For a 2D space with Cartesian coordinates $(x, y)$, the representation at a given point is a population vector $\mathbf{p}(x, y) \in \mathbb{R}^N$. Each of the $N$ components of this vector corresponds to the firing rate of a neuron, making the vector a point in an $N$-dimensional state space that captures the activity of the entire neural ensemble. Building upon previous modeling approaches Gao et al. (2021); McNamee et al. (2021); Xu et al. (2022), we define this map using

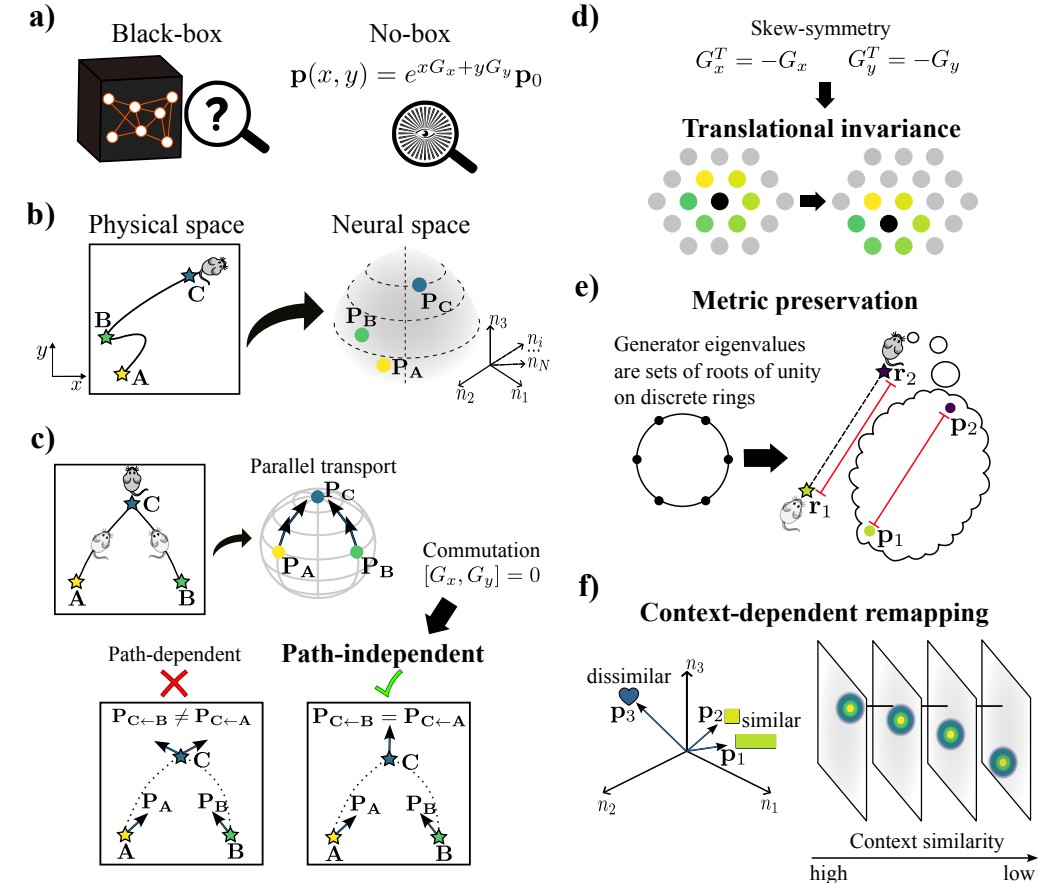

Figure 1: **The exponential map framework for interpretable spatial representations. a)** Deep learning models are "black-boxes" that learn spatial representations, but the underlying principles are obscured by the complexities of architecture, training, and objective functions. The exponential map model is a transparent "no-box" alternative, using generator matrices $(G_x, G_y)$ to construct a population vector $\mathbf{p}(x, y)$. **b)** A neural population vector, which captures the activity of the entire neural ensemble, is assigned to every location, mapping physical space to a neural representational space. **c) Path Invariance:** Path integration can be viewed as a form of parallel transport, where a vector representing the neural representation is moved along a trajectory in a high-dimensional state space. Traversing a curved manifold can induce a net transformation in the vector at a point that is dependent on the traversal route. By imposing simple, interpretable algebraic constraints on the model's generators, we can directly enforce fundamental properties. Path invariance is guaranteed if the generators commute. **d) Translational Invariance:** Making the generator matrices skew-symmetric ($G^T = -G$) imposes several biologically-relevant properties on the representation. First, it ensures that spatial relationships are consistently maintained across locations (translational invariance). Second, skew-symmetric generators produce orthogonal representations, meaning the population vector $\mathbf{p}(x, y)$ maintains a constant norm across the entire space. **e) Metric Preservation:** Preserving the geometry of flat space requires the generator eigenvalues to form sets of roots of unity on discrete rings in frequency space, which, for certain symmetry orders gives rise to grid-like patterns. **f) Remapping:** Generalizing the framework to non-spatial inputs allows for modeling remapping. The stacked sheets represent spatial maps for distinct values of a non-spatial context signal, $s$. As $s$ changes, the peak of activity shifts location even if the physical position $(x, y)$ is constant. This remapping preserves similarity: similar context values (neighboring sheets) result in spatially proximal firing fields.

the matrix exponential:

$$\mathbf{p}(x, y) = e^{xG_x + yG_y}\mathbf{p}_0, \tag{1}$$

where $G_x, G_y \in \mathbb{R}^{N \times N}$ are *generator matrices* for the cardinal directions and $\mathbf{p}_0 = \mathbf{p}(0,0)$ is the representation at some origin point. The generator matrices define how locations in physical space translate into transformations in the high-dimensional neural state space. The exponential map then composes these transformations to "transport" the origin vector, $\mathbf{p}_0$ to a population vector at any target location $(x, y)$.

While Eq. (1) provides a constructive method for generating a spatial map, without further constraints, an arbitrary choice of generators could produce a map ill-suited for navigation. For instance, the representation could prove trivial (all locations map to the same vector) or ambiguous (multiple locations map to the same vector). As we will show, the power of this framework lies in its transparency, allowing us to derive precise algebraic conditions on $G_x$ and $G_y$ that guarantee properties essential to navigation.

## 2.2 FROM SPATIAL REPRESENTATION TO PATH INTEGRATION AND PATH INDEPENDENCE

Path integration is a crucial skill possessed by most animals, wherein one's location is inferred by integrating past location and self-motion information. In the context of the representation defined in Eq. (1), path integration is realized if the representation at a new location can be derived from the current representation via a displacement operator $Q$:

$$\mathbf{p}(x + \Delta x, y + \Delta y) = Q(\Delta x, \Delta y)\mathbf{p}(x, y). \tag{2}$$

Intuitively, we can say that we can perform path integration, if, for any past location $(x, y)$ and the corresponding population vector $\mathbf{p}(x, y)$ we can arrive at the correct population vector $\mathbf{p}(x + \Delta x, y + \Delta y)$ at the new location $(x + \Delta x, y + \Delta y)$ through some operation $Q$ that only depends on the displacement. Inserting our spatial representation from Eq. (1), we find that we want

$$e^{(x+\Delta x)G_x + (y+\Delta y)G_y}\mathbf{p}_0 = Q(\Delta x, \Delta y)e^{xG_x + yG_y}\mathbf{p}_0.$$

This equality suggests that we want

$$Q(\Delta x, \Delta y) = e^{\Delta x G_x + \Delta y G_y}.$$

However, the exponential function in Eq. (1) is a matrix exponential, which behaves differently from the regular exponential function. In particular, the Baker-Campbell-Hausdorff formula dictates that

$$e^A e^B = e^{A + B + \frac{1}{2}[A,B] + \frac{1}{12}[A,[A,B]] + \frac{1}{12}[B,[B,A]] + \cdots},$$

where $[A, B] = AB - BA$ is the commutator between matrices $A$ and $B$. However, this immediately reveals that if the generator matrices $G_x$, $G_y$ commute, $[G_x, G_y] = 0$, then Eq. (2) is automatically satisfied for any displacement, as

$$[aG_x + bG_y, a'G_x + b'G_y] = (ab' - a'b)[G_x, G_y],$$

for all $(a, b)$ and $(a', b')$. Thus, if the generators commute, the model can path integrate exactly and indefinitely. An important effect of this choice is that the representation is path-invariant (as illustrated in Fig. 1c), meaning that the population vector at a point does not depend on the path taken to it. This is also demonstrated explicitly in Appendix B. Going forward, we therefore demand that $G_x$, $G_y$ commute, which ensures that the representation $\mathbf{p}$ is path-integration compatible, as enacted by Eq. (2). Next, we demonstrate that commuting generator matrices enable an explicit construction that allows us to specify the similarity structure of the spatial representation.

## 2.3 ORTHOGONAL TRANSFORMATIONS FOR EGOCENTRIC NAVIGATION

With a path integration-compatible model established, we turn to the properties of the representation itself. A critical metric for representational structure is the similarity between population vectors at distinct locations. Considering the path-integrating model defined in Eq. (2), we examine the normalized inner product (cosine similarity) between a population vector at $(x, y)$ and a target vector at $(x + \Delta x, y + \Delta y)$:

$$C(x, y, \Delta x, \Delta y) = \frac{\mathbf{p}(x, y)^T Q(\Delta x, \Delta y)\mathbf{p}(x, y)}{|\mathbf{p}(x, y)||Q(\Delta x, \Delta y)\mathbf{p}(x, y)|},$$

Substituting the exponential map form, using that $(e^A)^T = e^{A^T}$, and assuming commuting generators, the similarity expression transforms to:

$$C(x, y, \Delta x, \Delta y) = \mathbf{p}_0^T e^{x(G_x^T + G_x) + y(G_y^T + G_y) + \Delta x G_x + \Delta y G_y} \mathbf{p}_0 / Z, \tag{3}$$

where $Z$ is the normalization factor from before. Inspection of Eq. (3) reveals a fundamental geometric condition. If the generator matrices $G_x$ and $G_y$ are skew-symmetric ($G^T = -G$), two key properties emerge. First, the matrix exponential of a skew-symmetric matrix is orthogonal. This ensures that the transformation preserves the norm of the population vector everywhere ($|\mathbf{p}(x, y)| = |\mathbf{p}_0|$), fixing the normalization factor to $Z = |\mathbf{p}_0|^2$. Second, the position-dependent terms in the exponent vanish because $G^T + G = 0$, as illustrated in Fig. 1d). Consequently, the similarity becomes strictly translation invariant, depending solely on the displacement $(\Delta x, \Delta y)$. We therefore enforce the condition:

$$G_x^T = -G_x \quad G_y^T = -G_y,$$

which yields the displacement-dependent similarity:

$$C(\Delta x, \Delta y) = \frac{\mathbf{p}_0^T e^{\Delta x G_x + \Delta y G_y} \mathbf{p}_0}{|\mathbf{p}_0|^2}, \tag{4}$$

This property is functionally critical for navigation. In an open-field regime where no locations are inherently privileged, the similarity structure should remain consistent across the environment. Furthermore, it enables spatial inferences (such as distance estimation; see Section 2.4) without requiring absolute positional information, making orthogonal transformations an ideal basis for egocentric navigation.

Similarity translational invariance is a recurring feature in theoretical models of spatial coding. For instance, in Continuous Attractor Neural Networks (CANNs), the activity profile shifts across the neural sheet without changing shape, naturally preserving the similarity structure between displaced states (Burak & Fiete, 2009; McNaughton et al., 2006). Similarly, frameworks based on transition coding and the Successor Representation (SR) demonstrate that eigenvectors of the environment's transition matrix, which essentially encode transition probabilities, naturally capture spatial periodicities and translational symmetries in open environments (Stachenfeld et al., 2017; Waniek, 2018). More recently, sequence coding models have shown that encoding trajectories via path integration leads to conformal isometries, where displacements in neural space differ from physical displacements only by a scale factor (Waniek, 2020; RG et al., 2025). Our framework unifies these observations by deriving the property explicitly from the algebraic structure of the generator matrices. While previous approaches often obtain these representations through statistical learning of transitions or optimization of spatiotemporal sequences (Waniek, 2018), we identify the exact algebraic condition—skew-symmetry ($G^T = -G$)—that guarantees this property for high-dimensional vector representations. This condition ensures that the induced transformations are orthogonal, thereby maintaining a constant norm (an equinorm constraint) while rendering the similarity function purely dependent on displacement. This provides a rigorous algebraic description of the "shift-invariant" connectivity required by biological circuits, identifying skew-symmetry as the structural necessity for any path-integrating system that preserves representational similarity.

Given skew-symmetric generators $G_x$ and $G_y$, we can decompose them into a canonical block diagonal form via the spectral theorem:

$$G_x = R^T \Sigma_x R \quad \text{and} \quad G_y = R^T \Sigma_y R, \tag{5}$$

where $R \in \mathbb{R}^{N \times N}$ is an orthogonal matrix, and $\Sigma_x, \Sigma_y$ are real block diagonal matrices with $2 \times 2$ skew-symmetric blocks. Assuming $N$ is even, the non-zero entries of these blocks correspond to the imaginary parts of the eigenvalues, which appear in conjugate pairs $\pm(i\lambda_{i,x}, i\lambda_{i,y})$. This structure ensures that $G_x$ and $G_y$ commute, as the constituent $2 \times 2$ skew-symmetric blocks commute and $R^T R = I$. Under these conditions, the similarity function simplifies to:

$$C(\Delta x, \Delta y) = \sum_i^N \alpha_{0,i}^2 \cos(\lambda_{i,x} \Delta x + \lambda_{i,y} \Delta y), \tag{6}$$

where $\boldsymbol{\alpha}_0 = R \frac{\mathbf{p}_0}{|\mathbf{p}_0|}$ and $\lambda_{i,x}, \lambda_{i,y}$ are the imaginary parts of the $i$-th eigenvalues of $G_x$ and $G_y$, respectively (see Appendix E for the derivation).

The translational invariance derived here offers a significant functional advantage: the similarity depends only on the relative difference between inputs, not their absolute values. Consequently, the framework generalizes naturally to non-spatial variables. By introducing a context signal $s$ corresponding to its own generator, the system can model "contextual displacements" independent of spatial displacements. As we explore in Section 2.5 (and illustrate in Fig. 1f), this property allows a single model to generate distinct, orthogonal spatial maps for different environmental contexts, mimicking hippocampal remapping (Leutgeb et al., 2004; Fyhn et al., 2007), simply by shifting the effective origin of the representation.

## 2.4 PRESERVING THE METRIC OF FLAT SPACE

Given a spatial representation and a notion of representational similarity, we determine what properties the representation must possess to be geometrically consistent. Following Gao et al. (2021) and Xu et al. (2022), we posit that a foundational property of any spatial representation is the faithful translation of physical distances into distances on a neural manifold. In the open field, where all directions and locations are effectively equivalent, distances should not appear warped in any particular location or direction. Consequently, the metric induced by Eq. (1) must match the flat Euclidean metric, at least up to a constant factor (a condition known as conformal isometry (Xu et al., 2022)). Demanding metric preservation of a path-integrating, orthogonal representation, yields a simple condition on the spectra of the generators $G_x$ and $G_y$: the representation preserves the flat metric if the eigenvalues form sets of roots of unity (see Appendix D for details and Fig. 1e for an illustration). Concretely, expressed in polar coordinates as $\lambda_{i,x} = k_i \cos(\phi_i)$ and $\lambda_{i,y} = k_i \sin(\phi_i)$ (where $\lambda$ are the imaginary parts of the eigenvalues), a flat metric-preserving representation satisfies

$$\sum_j^{N/2} \rho_j^2 e^{2i\phi_j} = 0,$$

where $\rho_i = \alpha_i k_i$ is shared by conjugate eigenvalues (see Appendix D). Geometrically, this implies that the eigenvalue angles $\phi_i$ must be evenly spaced on discrete rings in the frequency domain. For a given ring of radius $\rho_m$ with symmetry order $M$, the eigenvalues are distributed as

$$\phi_i \to \varphi_i + \pi \frac{i}{M},$$

with $i = 0, 1, ..., M - 1$.

To visualize the spatial representation produced by a particular symmetry $M$, we note that for orthogonal matrices, the entries of the population vector $\mathbf{p}$ constitute superpositions of 2D plane waves, denoted $\boldsymbol{\alpha}$ (see Appendix C). The specific interference pattern is determined by the choice of the orthogonal matrix $R$, as $\mathbf{p}_0 = R^T \boldsymbol{\alpha}$. Figure 2 shows example representations $\mathbf{p}$ and plane waves $\boldsymbol{\alpha}$ for different symmetries $M$, using a randomly sampled $R$ (see Appendix A for details). For a single ring, lower-order symmetries such as $M = 2$ and 3 produce plane wave mixtures oriented at $90°$ and $60°$, respectively. This results in grid-like representations: square-type grids for $M = 2$ and hexagonal grids for $M = 3$. Higher-order symmetries, for example, $M = 4$, yield increasingly complex interference patterns, resembling the honeycomb-like structures recently observed in RNNs trained for dual-agent path integration (Redman et al., 2024). Notably, some representations may not appear purely grid-like due to the random mixing induced by $R$. As $M$ increases, the representation becomes heterogeneous and lacks obvious periodicity. However, while the spatial tuning curves are strongly influenced by $R$, the similarity function depends only on $\boldsymbol{\alpha}$. With increasing $M$, the similarity becomes approximately radial; for $M = 20$, the similarity function approximates a Bessel function, as predicted in Appendix F.

Beyond uncovering a general condition for metric preservation, we find that the admissible solutions support the modular organization of grid cells observed in the brain (Stensola et al., 2012). Analyzing the similarity function (see Appendix F) reveals that if modules with the same spacing form roots of unity in their orientation, and the symmetry of the module orientation is co-prime to that of the pattern, the representation becomes head-direction independent over a large spatial range. Furthermore, if the relative spacing of different modules is proportional to the zeros of the Bessel function $J_0$, the similarity function approximates a Fourier-Bessel series. This allows for the construction of radially symmetric similarity functions capable of tuning navigation to specific length scales. Intriguingly, the average ratio of successive low-order Bessel function zeros used to

construct such a series falls within the variability range of experimentally observed grid cell module spacings (approximately $\sqrt{2}$) (Stensola et al., 2012), suggesting a link between the roots-of-unity algebraic structure and the organization of the entorhinal cortex.

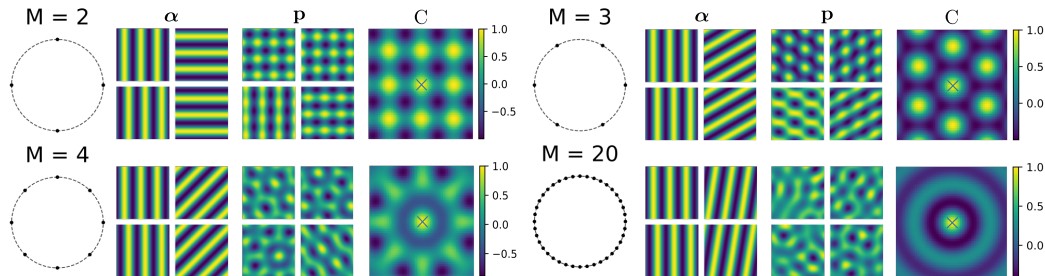

Figure 2: **Metric-preserving representations and roots of unity.** Example plane waves ($\boldsymbol{\alpha}$) and corresponding representations ($\mathbf{p}$) alongside the similarity function ($C$) relative to the origin (black cross) for representations whose eigenvalues form sets of roots of unity, with symmetries $M$ on a single ring. For each eigenvalue (imaginary part indicated by black dot), the corresponding conjugate eigenvalue is also shown. Representations were formed using generators with a single set root-of-unity solution with varying $M$, a random orthogonal matrix $R$, and $\mathbf{p}_0 = R^T \mathbf{1}$.

## 2.5 SIMILARITY PRESERVATION AND REMAPPING

With the generators defined in Eq. (5) and a choice of similarity function, we can generate spatial representations for a single environment up to a choice of orthogonal transformation $R$. However, animals are capable of distinctly encoding a variety of spatial and non-spatial information, such as room identity or olfactory cues, through remapping. In this section, we extend our model to this larger class of representations.

This generalization follows from the observation that between-representation similarities depend solely on the *spatial* displacement between them. If we encode non-spatial information while fixing the spatial location, representational similarities depend only on the change in the non-spatial input. We consider a global scalar signal $s$, such as a context variable, encoded identically to spatial coordinates:

$$\mathbf{p}(x, y, s) = e^{xG_x + yG_y + sG_s} \mathbf{p}_0. \tag{7}$$

The representation is coupled to the non-spatial signal via a generator matrix $G_s$. As with the spatial generators, we let $G_s = R^T \Sigma_s R$. The similarity between representations for two distinct context signals $s$ and $s'$ at a fixed location is then:

$$\mathbf{p}(x, y, s)^T \mathbf{p}(x, y, s') = \mathbf{p}^T e^{(s-s')G_R} \mathbf{p} = \mathbf{p}^T e^{\Delta s G_R} \mathbf{p}.$$

This inherits the form of the spatial similarity function. Consequently, comparing across contexts reveals that representations change even as spatial location remains fixed, mimicking the remapping behavior of spatial cells (Leutgeb et al., 2004; Fyhn et al., 2007).

Encoding non-metric information, such as context, raises a distinct challenge: unlike physical space, there is no intrinsic metric to preserve. Instead, we require that similar context signals produce similar representations, while dissimilar contexts result in orthogonal ones. Prior studies have demonstrated that such similarity preservation yields localized receptive fields, resembling biological place fields, when applied to spatial inputs (Sengupta et al., 2018; Pettersen et al., 2024). To implement this algebraically, we observe that the similarity function may be written as:

$$C(\Delta s) = \sum_i \alpha_{0,i}^2 \cos(\Delta s \lambda_{i,s}), \tag{8}$$

where $\lambda_{i,s}$ denotes the imaginary part of the $i$th eigenvalue of $G_s$ (assuming fixed spatial location). Since this expression is derived from the cosine similarity, it is strictly bounded to the interval $[-1, 1]$. To achieve similarity preservation, we seek a profile $C(\Delta s)$ that decays with increasing

$\Delta s$ towards a baseline level where inputs are deemed dissimilar. We can approximate a wide class of such functions by recognizing that Eq. (8) is effectively a cosine series with non-negative coefficients. Specifically, it may be viewed as a discrete approximation of the inverse Fourier transform of a symmetric function with a non-negative Fourier spectrum:

$$f(x) = \frac{1}{\sqrt{2\pi}} \int_{-\infty}^{\infty} F(k)e^{ikx}dk = \frac{1}{\sqrt{2\pi}} \int_{-\infty}^{\infty} F(k)\cos(kx)dk$$

$$\approx \frac{1}{\sqrt{2\pi}N} \sum_{i}^{N} \frac{F(k_i)}{p(k_i)} \cos(k_i x),$$

where the summation represents a Monte Carlo estimate of the integral using importance sampling with density $p(k_i)$.

For example, to approximate a Gaussian similarity function $f(x) = e^{-\sigma^2 x^2}$ (which ensures that dissimilar contexts become decorrelated), we sample the eigenvalues $\lambda_{i,s}$ from the Fourier transform of the Gaussian, that is, a normal distribution $\mathcal{N}(0, 2\sigma^2)$. Setting the coefficients $\alpha_{0,i}^2 = 1/N$ (achieved if $\mathbf{p}_0 = \frac{1}{\sqrt{N}} R^T \mathbf{1}$) yields:

$$C(\Delta s) \approx e^{-\sigma^2 \Delta s^2}.$$

Going forward, we set $\sigma = 1$.

To demonstrate this remapping behavior, we instantiated a metric-preserving spatial representation consisting of 10 sets of identical root-of-unity solutions and extended it to encode a non-spatial signal $s$ via Eq. (7). Multiple sets were used to ensure the Monte Carlo estimate approximates the Gaussian sufficiently well. The resulting between-context similarity is shown in Fig. 3. Similarities decay with increasing context dissimilarity, and the rate maps of unit activity shift between contexts, reproducing remapping dynamics (Fyhn et al., 2007). Crucially, spatial similarities are preserved for any fixed $s$, maintaining the grid-like structure derived in the previous section.

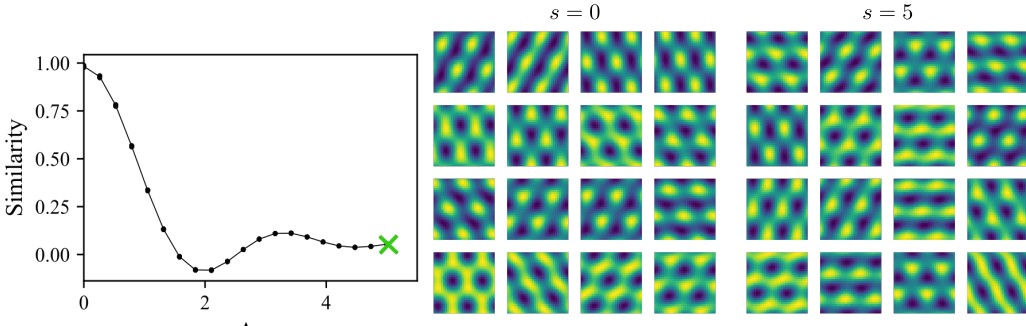

Figure 3: **Context-dependent remapping and similarity preservation.** Between-context similarity as a function of context separation. Also inset are example rate maps for two distant context values ($s = 0$, $s = 5$) corresponding to $\Delta s = 5$. Representations were formed using generators with 10 identical root-of-unity solutions with $M = 3$, a random orthogonal matrix $R$, and $\mathbf{p}_0 = R^T \mathbf{1}$.

Finally, we find that relaxing the metric preservation requirement in the spatial domain, demanding only similarity preservation via Gaussian-sampled eigenvalues for $G_x$ and $G_y$, results in an approximate Gaussian spatial similarity (see Appendix G). In this regime, spatial representations become heterogeneous and localized, resembling hippocampal place fields (O'Keefe & Dostrovsky, 1971). Thus, by modulating the similarity function, the exponential map model can generate the diverse range of spatial tuning curves observed in the brain. Notably, these spatial tuning curves are compatible with the Probabilistic Population Codes (PPC) framework (Ma et al., 2006; Beck et al., 2008). Specifically, if neural variability follows a distribution in the exponential family, for example, Poisson-like, with linear sufficient statistics, the Gaussian-like tuning curves derived here can support optimal Bayesian inference via linear integration. In this context, the algebraically derived representations effectively define the tuning kernel within the PPC formalism.

## 3 Biological Realization and Functional Navigation

While derived from algebraic principles, the exponential map framework maps directly onto biological mechanisms. We demonstrate that our model emerges from the dynamics of Continuous Attractor Neural Networks (CANNs) (Appendix H), can learn and even generalize experimental data (Appendix I), and supports robust, interpretable, multimap goal-oriented navigation (Appendix J).

### 3.1 Emergence from Attractor Dynamics and Local Learning

The exponential map model emerges naturally as the effective on-manifold dynamics of a gain-modulated CANN flowing towards a hypersphere attractor (see Appendix H.1). By analyzing the Lyapunov energy of the network, we find that the dynamics decompose into a stabilizing non-linear term, which constrains activity to the manifold, and a linear transport term. On the manifold, the state evolves according to $\dot{\mathbf{z}} = U(\mathbf{v})\mathbf{z}$, where $U$ is the velocity-dependent skew-symmetric component of the recurrent weights. This formulation physically identifies our algebraic generator matrices with the synaptic connectivity of the circuit. Consequently, the algebraic constraints derived in this work map directly to biological connectivity patterns. The skew-symmetry condition, required for translational invariance, corresponds to the asymmetric component of the recurrent weights. The commutativity requirement ensures that the time-ordered integration of synaptic inputs simplifies to a state-independent update (a vanishing Magnus expansion), providing a dynamical definition of path integration (see Appendix H.2). Thus, the exponential map offers a rigorous description of how recurrent networks perform exact temporal integration without trajectory-dependent errors. Furthermore, we show that such connectivity matrices need not be hard-coded; rather, they can emerge naturally via a biologically plausible local learning rule that exploits time-lagged anti-symmetric correlations. Finally, we establish that for hexagonal grid cells, the grid spacing scales linearly with the attractor network's time constant. This finding offers a normative explanation for the grid-scale hierarchy observed along the dorsal-ventral axis of the medial entorhinal cortex (MEC) (see Appendix H.3).

### 3.2 Learning from Data and Functional Navigation

To validate the generative ability of the framework, we trained the model to reproduce experimental grid cell rate maps by minimizing reconstruction error subject to a commutation penalty (see Appendix I). The learned model successfully extrapolates grid patterns beyond the training boundaries, demonstrating that it captures the intrinsic algebraic structure underlying the biological data.

Functionally, the framework enables robust navigation by leveraging principles from Hyperdimensional Computing (HDC) (see Appendix J). Salient locations can be aggregated via "bundling" into memory vectors $\mathbf{p}_R = \sum \mathbf{p}(x_i, y_i)$, creating a similarity landscape that supports navigation via gradient ascent. Furthermore, context-dependent remapping serves as a "binding" operation, effectively orthogonalizing representations across contexts. This allows multiple reward maps to be superimposed within a single neural population, facilitating context-specific retrieval and flexible goal-oriented navigation.

## 4 Conclusion

In this work, we introduced a first-principles framework for generating neural spatial representations using an exponential map model. By leveraging generator matrices and the matrix exponential, we bypassed the "black-box" nature of deep learning models, enabling a transparent and theoretically-grounded investigation into the principles of neural navigation. We derived the exact algebraic conditions required for a coherent map of space. Specifically, we demonstrated that commuting generators are necessary to guarantee path-independent representations, a critical requirement for accurate path integration. Furthermore, we showed that constraining generators to be skew-symmetric produces orthogonal transformations, yielding representations with translationally invariant similarity structures, an ideal property for egocentric navigation in open-field environments. We also established that preserving the flat metric of Euclidean space requires the generator eigenvalues to form sets of roots of unity on discrete rings in the frequency domain. Despite its mathematical simplicity, the proposed framework constructs a diverse range of biologically plausible spatial tuning curves,

including grid cells and place cells, and models context-dependent remapping by extending these principles to non-spatial inputs. This work offers an interpretable alternative to conventional deep learning approaches, revealing the fundamental mathematical structures that may underpin how the brain represents and navigates through space.

## 5  LIMITATIONS AND FUTURE WORK

While our framework provides a transparent account of how coherent spatial maps can be formed, it has several limitations that open avenues for future research.

First, the current model is primarily developed for navigation in flat, open-field environments. Animals, however, must navigate complex, curved, and obstacle-laden spaces. Future work should explore how the generator framework can be extended to represent non-Euclidean geometries. This may involve introducing position-dependent or non-commuting generators that reflect the local topology and geometry of the environment.

Second, our remapping model currently uses a scalar context signal. Generalizing this to handle high-dimensional, structured inputs, such as visual scenes or complex sensory cues, is a critical next step. The algebraic structure naturally supports vector-valued generators ($\mathbf{s} \cdot \mathbf{G}_s$); in this regime, similarity preservation implies preserving the semantic distances between high-dimensional inputs, allowing for the modeling of how environmental identity and spatial location are integrated into a unified representation. This would bridge the gap between our algebraic approach and the rich, multi-modal inputs processed by biological and artificial systems.

Third, while geometric considerations fix most model parameters, the choice of the orthogonal matrix $R$ remains a degree of freedom. In this work, we restricted our analysis to randomly sampled matrices, which strongly influence the resulting tuning curves by mixing the underlying plane waves. Notably, this choice can be dissociated from the representational similarity structure: as shown in the remapping analysis, selecting an appropriate initial vector $\mathbf{p}_0$ renders the similarity function $C$ independent of $R$. This suggests that while individual tuning curves depend on $R$, the overall geometry of the neural map does not. Future work should investigate whether biological constraints—such as metabolic energy efficiency (Cueva & Wei, 2018), non-negativity (Sorscher et al., 2023), or extrinsic distance preservation (Xu et al., 2025)—mandate specific matrices $R$. In particular, Xu et al. (2025) demonstrated that hexagonal symmetry ($M = 3$) is optimal for preserving extrinsic distances (global Euclidean distance) in periodic representations, potentially explaining the prevalence of hexagonal grids over square lattices. Additionally, improved quadrature rules for similarity function approximation warrant exploration. For instance, the eigenfunctions of the Laplacian can construct optimal truncated Fourier series (Bronstein et al., 2021), suggesting that coefficient selection strategies beyond our Monte Carlo approach may yield superior approximations, while also selecting for periodic solutions. Together, these constraints could drive generated representations toward the specific hexagonal or sparse place-bound tunings observed in the brain.

Finally, while we propose exact algebraic conditions for properties like path integration and metric preservation, our framework is primarily descriptive rather than prescriptive regarding their acquisition. Although we have outlined a local learning mechanism for the emergence of skew-symmetric weights in Appendix H, a critical direction is to extend the framework to incorporate biological noise and imperfect commutation. Future research should focus on developing learning rules that yield approximately commuting weight matrices from realistic synaptic plasticity, while quantifying how deviations from perfect commutation accumulate into path integration errors. This would bridge the gap between the mathematical idealization and noisy neural circuits, establishing tolerance bounds for biological navigation systems.

## 6  CODE AVAILABILITY AND DISCLOSURES

All code used to generate the results and figures in this work will be made available upon publication.

Large language models were used in writing this paper, with usage limited to improving writing and readability.

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

APPENDIX

# A    METHODS

All computational simulations were implemented using the matrix exponential operator in PyTorch (Paszke et al., 2019). Random orthogonal matrices were generated using the `ortho_group` module from the SciPy library (Virtanen et al., 2020), which samples uniformly from the orthogonal group $O(N)$. For experiments involving root-of-unity solutions, the spatial domain was set to $20 \times 20$ to capture the full periodicity of the representation. For the similarity-preserving experiments, the domains were defined as $s \in [0, 5]$ for the non-spatial context signal and $(x, y) \in [-2, 2]$ for the spatial coordinates, commensurate with the length scale of the target Gaussian similarity function.

# B    COMMUTING GENERATORS PRODUCE PATH-INDEPENDENT
##       REPRESENTATIONS

Since the spatial representation is defined by the exponential map in Eq. (1), we can enforce specific geometric properties by constraining the generators $G_x$ and $G_y$. For example, Schaeffer et al. (2023)proposed that representations should be path-independent, that is, the representation at a given location must not depend on the trajectory taken to reach it. In the exponential map formalism, this condition is satisfied generators commute. To illustrate this, consider two distinct paths to a point $D$ starting from $A$: a path $A \to B \to D$ and a path $A \to C \to D$. The resulting representations are generated by composing the transformations for each segment:

$$\mathbf{p}_{ABD} = e^{\Delta x_{BD} G_x + \Delta y_{BD} G_y} e^{\Delta x_{AB} G_x + \Delta y_{AB} G_y} \mathbf{p}_A,$$
$$\mathbf{p}_{ACD} = e^{\Delta x_{CD} G_x + \Delta y_{CD} G_y} e^{\Delta x_{AC} G_x + \Delta y_{AC} G_y} \mathbf{p}_A,$$

where $\mathbf{p}_A$ denotes the representation at $A$. If $G_x$ and $G_y$ commute, their linear combinations also commute. By the Baker-Campbell-Hausdorff (BCH) formula, the composite transformations then combine into a single exponential summing the exponents:

$$\mathbf{p}_{ABD} = e^{(\Delta x_{AB} + \Delta x_{BD}) G_x + (\Delta y_{AB} + \Delta y_{BD}) G_y} \mathbf{p}_A = \mathbf{p}_{ACD}.$$

Here, the final state depends only on the net displacement from the initial location, which is identical for both paths.

Conversely, if the generators do not commute, path dependence arises from non-vanishing terms in the BCH expansion. For matrices $U$ and $V$, the expansion is given by:

$$e^U e^V = e^{U + V + \frac{1}{2}[U,V] + \frac{1}{12}([U,[U,V]] - [V,[U,V]]) + \cdots}.$$

Applying this to the path $A \to B \to D$, we define $U = \Delta x_{AB} G_x + \Delta y_{AB} G_y$ and $V = \Delta x_{BD} G_x + \Delta y_{BD} G_y$. The representation becomes:

$$\mathbf{p}_{ABD} = e^U e^V \mathbf{p}_A = e^{U + V + \frac{1}{2}[U,V] + \cdots} \mathbf{p}_A.$$

The commutator $[U, V]$ expands to:

$$[U, V] = [\Delta x_{AB} G_x + \Delta y_{AB} G_y, \Delta x_{BD} G_x + \Delta y_{BD} G_y].$$

Using the linearity of the commutator and the property $[G_x, G_y] = -[G_y, G_x]$, this simplifies to:

$$[U, V] = (\Delta x_{AB} \Delta y_{BD} - \Delta y_{AB} \Delta x_{BD})[G_x, G_y].$$

Crucially, if $[G_x, G_y] \neq 0$, the exponent includes a term proportional to the cross product of the path segment displacements (geometrically related to the area enclosed by the path components).

Similarly, for the path $A \to C \to D$ with operators $W$ and $Z$, the commutator $[W, Z]$ introduces different displacement cross-terms. Consequently, the higher-order corrections differ between the two paths, and $\mathbf{p}_{ABD} \neq \mathbf{p}_{ACD}$. Thus, commutativity of the generators is a necessary condition for the representation to depend solely on the net displacement.

## C  FROM GENERATORS TO REPRESENTATIONS

The structure of the spatial representation in Eq. (1) allows for an explicit decomposition of individual cell responses. Assuming unit-norm $\mathbf{p}_0$ and commuting, skew-symmetric generators in block-diagonal form (as in Eq. (5)), the dynamics in the rotated basis $\boldsymbol{\alpha} = R\mathbf{p}$ are given by:

$$\boldsymbol{\alpha}(x,y) = e^{x\Sigma_x + y\Sigma_y}\boldsymbol{\alpha}_0,$$

where $\boldsymbol{\alpha}_0 = R\mathbf{p}_0$. In this basis, the matrix exponential reduces to a block-diagonal matrix populated by $2 \times 2$ rotation matrices. Following Dorrell et al. (2023), the representation decomposes into independent rotations within distinct 2D subspaces. For the $i$-th block ($i = 1, \ldots, N/2$), the update rule is:

$$\boldsymbol{\alpha}^i = \begin{pmatrix} \cos(\Omega_i) & -\sin(\Omega_i) \\ \sin(\Omega_i) & \cos(\Omega_i) \end{pmatrix} \boldsymbol{\alpha}_0^i$$

where $\boldsymbol{\alpha}^i \in \mathbb{R}^2$ represents the state vector within the $i$-th subspace. The rotation angle $\Omega_i = x\lambda_{i,x} + y\lambda_{i,y}$ couples the spatial displacement to the generator eigenvalues.

Letting $\alpha_1^i$ and $\alpha_2^i$ denote the two components of the $i$-th block (corresponding to the conjugate eigenvalue pair), the matrix multiplication yields:

$$\alpha_1^i = \alpha_{0,1}^i \cos(\Omega_i) - \alpha_{0,2}^i \sin(\Omega_i),$$
$$\alpha_2^i = \alpha_{0,1}^i \sin(\Omega_i) + \alpha_{0,2}^i \cos(\Omega_i).$$

By applying the harmonic addition theorem, these components can be rewritten as phase-shifted sinusoids:

$$\alpha_1^i = A_i \cos(x\lambda_{i,x} + y\lambda_{i,y} + \omega_i),$$
$$\alpha_2^i = A_i \sin(x\lambda_{i,x} + y\lambda_{i,y} + \omega_i),$$

where the amplitude is $A_i = \sqrt{(\alpha_{0,1}^i)^2 + (\alpha_{0,2}^i)^2}$ and the phase is $\omega_i = \arctan(\alpha_{0,2}^i/\alpha_{0,1}^i)$.

These equations demonstrate that, in the canonical basis, each component acts as a 2D plane wave with orientation and frequency determined by $\lambda_{i,x}$ and $\lambda_{i,y}$, and a phase shift $\omega_i$ along the wave direction. Since the observed neural representation is given by $\mathbf{p} = R^T\boldsymbol{\alpha}$, the firing rate of each neuron consists of a linear superposition (mixture) of these plane waves.

## D  METRIC PRESERVATION

Consider the representation along a parametrized trajectory $\mathbf{r}(t) = (x(t), y(t))$:

$$\mathbf{p}(x(t), y(t)) = e^{x(t)G_x + y(t)G_y}\mathbf{p}_0.$$

The length of this trajectory in the representational space is given by the path integral of the line element $ds = |d\mathbf{p}|$. By the chain rule, the differential change in the representation is:

$$d\mathbf{p} = \left( \frac{\partial \mathbf{p}}{\partial x}\frac{dx}{dt} + \frac{\partial \mathbf{p}}{\partial y}\frac{dy}{dt} \right) dt$$

The trajectory length $L$ can be expressed in terms of the induced metric $g_{ij}$ as:

$$L = \int_0^S \sqrt{|d\mathbf{p}|^2} = \int_0^T \sqrt{\sum_{ij} g_{ij}\frac{dr_i}{dt}\frac{dr_j}{dt}}\, dt.$$

Comparing with the squared line element, we can then simply read off the induced metric $g$ induced metric, as

$$g = -\begin{pmatrix} \mathbf{p}_0^T G_x^2 \mathbf{p}_0 & \mathbf{p}_0^T G_x G_y \mathbf{p}_0 \\ \mathbf{p}_0^T G_x G_y \mathbf{p}_0 & \mathbf{p}_0^T G_y^2 \mathbf{p}_0 \end{pmatrix},$$

where we have used the fact that $\mathbf{p}^T G_{r_i}^T G_{r_j} \mathbf{p} = -\mathbf{p}_0^T G_{r_i} G_{r_j} \mathbf{p}_0$ due to the skew-symmetry of the generator matrices.

We further simplify this using the block-diagonal decomposition $G = R^T \Sigma R$. Since $\Sigma$ contains $2 \times 2$ skew-symmetric blocks, $G_x^2 = R^T D_x R$, where $D_x$ is a diagonal matrix containing the squared imaginary part of a given eigenvalue $-\lambda_{i,x}^2$. The mixed term $G_x G_y$ similarly diagonalizes to entries $-\lambda_{i,x}\lambda_{i,y}$. Letting $\boldsymbol{\alpha}_0 = R\mathbf{p}_0$, the metric becomes:

$$g = -\begin{pmatrix} \boldsymbol{\alpha}_0^T D_x \boldsymbol{\alpha}_0 & \boldsymbol{\alpha}_0^T D_{xy} \boldsymbol{\alpha}_0 \\ \boldsymbol{\alpha}_0^T D_{xy} \boldsymbol{\alpha}_0 & \boldsymbol{\alpha}_0^T D_y \boldsymbol{\alpha}_0 \end{pmatrix}.$$

Substituting this back into the path length integral yields:

$$L = \int_0^T \sqrt{\sum_{i=1}^N \alpha_{0,i}^2 (\lambda_{i,x}^2 \dot{x}^2 + 2\lambda_{i,x}\lambda_{i,y}\dot{x}\dot{y} + \lambda_{i,y}^2 \dot{y}^2)}\, dt$$

To preserve the flat Euclidean metric (that is, $g = \sigma^2 I$), we require the off-diagonal terms to vanish and the diagonal terms to be equal:

$$\sum_{i=1}^N \alpha_{0,i}^2 \lambda_{ix}^2 = \sum_{i=1}^N \alpha_{0,i}^2 \lambda_{iy}^2, \quad \text{and} \quad \sum_{i=1}^N \alpha_{0,i}^2 \lambda_{ix}\lambda_{iy} = 0.$$

Introducing polar coordinates for the eigenvalues $\lambda_{i,x} = k_i \cos\phi_i$ and $\lambda_{i,y} = k_i \sin\phi_i$, and defining $\rho_i = \alpha_{0,i} k_i$, these conditions become:

$$\sum_{i=1}^N \rho_i^2 \cos^2(\phi_i) = \sum_{i=1}^N \rho_i^2 \sin^2(\phi_i)$$

$$\sum_{i=1}^N \rho_i^2 \cos\phi_i \sin\phi_i = 0.$$

Using trigonometric identities, this system simplifies to requiring that the weighted sum of phasors vanishes at double the angle:

$$\sum_{i=1}^N \rho_i^2 \cos(2\phi_i) = 0 \quad \text{and} \quad \sum_{i=1}^N \rho_i^2 \sin(2\phi_i) = 0,$$

which is equivalent to the complex condition

$$\sum_{i=1}^N \rho_i^2 e^{2i\phi_i} = 0.$$

Since eigenvalues appear in conjugate pairs ($\phi_j^* = \phi_j + \pi$), and $e^{2i(\phi+\pi)} = e^{2i\phi}$, the sum effectively runs over $N/2$ independent pairs. Assuming the simplest case, with equal weighting $\rho_i = \rho$, the condition reduces to finding a set of angles such that:

$$Z = \sum_{j=1}^{N/2} e^{2i\phi_j} = 0.$$

This condition is satisfied if the $N/2$ eigenvalue pairs are partitioned into subsets (modules), where each subset forms a collection of *roots of unity* that sums to zero. Specifically, for a single module of symmetry order $M$, we require $M$ eigenvalue pairs (consuming $2M$ dimensions of the total $N$) with angles distributed uniformly on the circle:

$$\phi_j = \pi \frac{j}{M}, \quad j = 0, 1, \dots, M-1.$$

Consequently, the full high-dimensional representation can be constructed as a linear combination of such sets. For a system with multiple modules $k = 1 \dots K$, each with radius $\rho_k$, symmetry $M_k$, and orientation $\varphi_k$, the total sum vanishes if each module vanishes individually:

$$Z = \sum_{j=1}^J \rho_j^2 e^{2i\varphi_j} \sum_{m=0}^{M_j-1} e^{2\pi i \frac{m}{M_j}}.$$

In other words, for each radius $\rho$, there can be multiple rotated sets of roots of unity, each with its own rotational symmetry.

A comparison with the explicit form of the representation in Appendix C reveals a striking parallel to the modular organization of grid cells in the medial entorhinal cortex (Hafting et al., 2005; Stensola et al., 2012), which are similarly organized into modules defined by grid spacing ($\rho$), orientation ($\varphi$), and symmetry ($M$).

## E   SIMILARITY FUNCTION DERIVATION

We derive the explicit form of the representational similarity starting from the Eq. (4), We assume the generators are skew-symmetric and commute, admitting the decomposition

$$G_x = R^T \Sigma_x R \quad \text{and} \quad G_y = R^T \Sigma_y R,$$

where $R$ is a shared orthogonal matrix. Using the identity $e^{P^{-1}AP} = P^{-1}e^A P$, we rewrite the similarity expression as

$$C(\Delta x, \Delta y) = \left( R \frac{\mathbf{p}_0}{|\mathbf{p}_0|} \right)^T e^{\Delta x \Sigma_x + \Delta y \Sigma_y} \left( R \frac{\mathbf{p}_0}{|\mathbf{p}_0|} \right)$$
$$= \boldsymbol{\alpha}_0^T e^{\Delta x \Sigma_x + \Delta y \Sigma_y} \boldsymbol{\alpha}_0,$$

where we define the rotated unit vector $\boldsymbol{\alpha}_0 \equiv R \frac{\mathbf{p}_0}{|\mathbf{p}_0|}$. Note that $\sum_i \alpha_{0,i}^2 = 1$ due to the orthogonality of $R$.

The exponent matrix $\Omega = \Delta x \Sigma_x + \Delta y \Sigma_y$ retains the block-diagonal, skew-symmetric structure of the generators. In the power series expansion of the matrix exponential, even powers $\Omega^{2n}$ result in diagonal matrices (as the square of a $2 \times 2$ skew-symmetric block is diagonal), while odd powers $\Omega^{2n+1}$ remain skew-symmetric. Since a quadratic form $\mathbf{x}^T A \mathbf{x}$ vanishes for any skew-symmetric matrix $A$, odd terms do not contribute to the similarity. The expansion therefore reduces to a sum over even powers:

$$C(\Delta x, \Delta y) = \boldsymbol{\alpha}_0^T \left( \sum_n^\infty \frac{(-1)^n}{(2n)!} D^{2n} \right) \boldsymbol{\alpha}_0,$$

where $D$ is a diagonal matrix with entries $\theta_i = \lambda_{i,x} \Delta x + \lambda_{i,y} \Delta y$, and $\lambda_{i,\cdot}$ denotes the imaginary part of the corresponding eigenvalue. Recognizing the Taylor series for the cosine function, the matrix sum converges to a diagonal matrix with entries $\cos(\theta_i)$. Consequently, the similarity simplifies to:

$$C(\Delta x, \Delta y) = \sum_i^N \alpha_{0,i}^2 \cos(\lambda_{i,x} \Delta x + \lambda_{i,y} \Delta y).$$

## F   DESIGNING SPATIAL SIMILARITY FUNCTIONS

We established that the similarity function Eq. (6) takes the general form of a weighted sum of cosines. To understand the structure of the resulting similarity function, we rewrite the expression in polar coordinates using $x = r \cos\theta$, $y = r \sin\theta$, and $\lambda_{i,x} = k_i \cos\phi_i$, $\lambda_{i,y} = k_i \sin\phi_i$:

$$C = \sum_i \alpha_{0,i}^2 \cos(k_i r \cos(\theta - \phi_i)),$$

Applying the Jacobi-Anger expansion

$$\cos(z \cos(\omega)) = J_0(z) + 2 \sum_{n=1}^\infty (-1)^n J_{2n}(z) \cos(2n\omega)$$

$$= J_0(z) + 2 \sum_{n=1}^\infty (-1)^n \Re \left\{ J_{2n}(z) e^{2in\omega} \right\},$$

where $J_n(z)$ is the $n$-th Bessel function of the first kind, we obtain:

$$C(r,\theta) = \sum_j \alpha_{0,j}^2 J_0(k_j r) + 2 \sum_{n=1}^{\infty} (-1)^n \Re \left\{ e^{2in\theta} \sum_j \alpha_{0,j}^2 J_{2n}(k_j r) e^{2in\phi_j} \right\}.$$

This decomposes the similarity into a purely radial component (the $J_0$ term) and a mixed term dependent on head direction $\theta$.

Further simplification relies on the structure of the eigenvalues. If the representation preserves the flat metric (see Appendix D), the eigenvalues form discrete roots-of-unity constellations. Assuming constant weighting $\alpha_{0,j}$ on each ring $j$, the inner phasor sum becomes:

$$\sum_j \alpha_{0,j}^2 J_{2n}(k_j r) e^{2in\phi_j} = \sum_j \alpha_{0,j}^2 J_{2n}(k_j r) \, e^{2in\varphi_j} \sum_{m=0}^{M_j-1} e^{2\pi imn/M_j},$$

where the geometric series $\sum_{m=0}^{M_j-1} e^{2\pi imn/M_j}$ vanishes unless $n$ is a multiple of the symmetry order $M_j$. Consequently, angular dependence only arises at harmonic orders $n = \ell M_j$. For large $M$, the similarity function becomes approximately isotropic (head-direction independent).

We can further suppress low-order angular terms by requiring the orientation offsets $\varphi_j$ of different modules to also form a root-of-unity constellation. Specifically, if we sum over a set of orientations indexed by $l$ such that:

$$\sum_j \alpha_{0,j}^2 J_{2n}(k_j r) e^{2in\phi_j} = \sum_j \alpha_{0,j}^2 J_{2n}(k_j r) \sum_l e^{2in\varphi_l} \sum_{m=0}^{M-1} e^{2\pi imn/M},$$

and the orientations satisfy the condition $\sum_l e^{2in\varphi_l} = \sum_{l=0}^{N-1} e^{2\pi iln/N}$, then non-zero terms persist only when $n$ is a multiple of $N$ and simultaneously a multiple of $M$. If $M$ and $N$ are coprime, the lowest order angular dependence is pushed to $n = MN$. In this case, the full similarity function is:

$$C(r,\theta) = \sum_j \alpha_{0,j}^2 J_0(k_j r) + 2MN \sum_{\ell=1}^{\infty} (-1)^{\ell NM} \alpha_{0,j}^2 J_{2\ell MN}(k_j r) \cos(2\ell MN\theta).$$

The magnitude of the angular terms is governed by the high-order Bessel functions. For small arguments $z$, the Bessel function of order $\gamma$ behaves as:

$$J_\gamma(z) \approx \frac{1}{\Gamma(\gamma+1)} \left(\frac{z}{2}\right)^\gamma$$

Here, the order is $\gamma = 2\ell MN$. For large $MN$, this term decays rapidly near the origin ($z\sqrt{\gamma+1}$). Consequently, for a large range of displacements $r$, the angular terms vanish, and the similarity becomes effectively radial:

$$C(r,\theta) \approx \sum_j \alpha_{0,j}^2 J_0(k_j r).$$

Intriguingly, this connection offers a theoretical prediction for grid spacing ratios. The ratio of successive zeros of the Bessel function $J_0$ converges toward an average value close to $\sqrt{2}$ when including low-order zeros (Fig. 4). This falls precisely within the variability range of grid module spacing ratios observed experimentally (Stensola et al., 2012). While grid scale ratios are often assumed to be $\sqrt{2}$ to maximize spatial range, our framework suggests they may arise from the optimal approximation of a radial similarity kernel via a Fourier-Bessel expansion. Finally, we note that while this ratio determines relative scaling, the Fourier-Bessel series inherently defines an absolute length scale related to the domain of approximation. This suggests that the absolute grid scale may be set by the size of the region the animal needs to encode reliably.

## G  SIMILARITY-PRESERVING SPATIAL REPRESENTATIONS

By relaxing the strict requirement for metric preservation, we can explore representations designed to preserve similarity structure. Following the approach outlined in Section 2.5, we consider the case

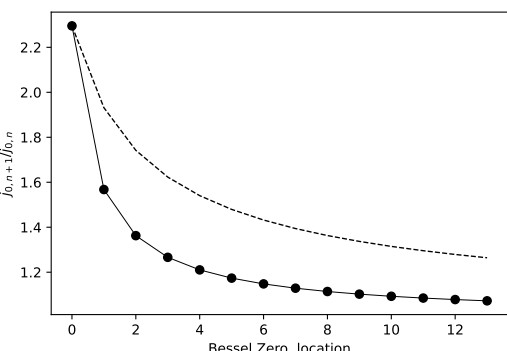

Figure 4: **Bessel function zeros and grid scale ratios.** Ratio of subsequent zeros of the Bessel function $J_0$ (large dots), alongside a cumulative average (dashed line).

where generator eigenvalues are sampled from a normal distribution. This choice yields an approximate Gaussian similarity function. To demonstrate that this generalizes to spatial representations, we simulated a population of $N = 256$ units with generator eigenvalues for $G_x$ and $G_y$ sampled from $\mathcal{N}(0, 2)$. The results are shown in Fig. 5. Unlike the periodic, grid-like patterns characteristic of low-order roots-of-unity solutions (metric preservation), these units exhibit heterogeneous, spatially localized tuning curves reminiscent of hippocampal place fields (O'Keefe & Dostrovsky, 1971). Given that place cells are known to encode both spatial and non-spatial cues, such as olfactory context, (Anderson & Jeffery, 2003), this result suggests that the context-dependent model in Eq. (7) could be naturally extended to model conjunctive representations of space and context, consistent with recent theoretical proposals (Pettersen et al., 2024).

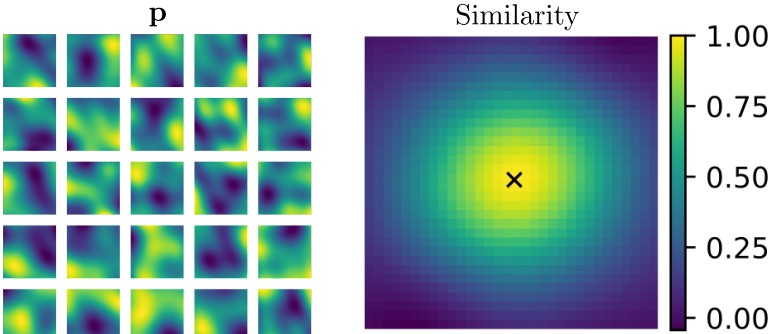

Figure 5: **Emergence of place-like fields from similarity preservation. Left:** Example rate maps for a model where generator eigenvalues are sampled from a normal distribution, resulting in an approximately Gaussian similarity function. **Right:** The resulting spatial similarity function relative to the origin.

## H   BIOLOGICAL INTERPRETATION

The exponential map framework provides a rigorous algebraic description of spatial representations. Here, we demonstrate that this framework is not merely an abstraction but emerges as the on-manifold dynamics of a gain-modulated Continuous Attractor Neural Network (CANN).

## H.1 Continuous Attractor Formulation

We consider the general dynamics of a recurrent neural network Zhang (1996); Ocko et al. (2018) governed by:

$$\tau \frac{d\mathbf{z}}{dt} = W[\mathbf{v}(t)]\sigma(\mathbf{z}(t)), \tag{9}$$

where $\mathbf{z} \in \mathbb{R}^N$ is the neural state vector and $\tau$ is the effective time constant. Here, $W[\mathbf{v}]$ represents the effective state-transition matrix modulated by an external input $\mathbf{v}$, such as 2D velocity, and $\sigma$ is a generalized non-linear function. Note that this formulation subsumes standard leaky-integrator models (for example, $\tau\dot{\mathbf{z}} = -\mathbf{z} + J\phi(\mathbf{z})$) if the decay term is absorbed into the effective interaction $W\sigma(\mathbf{z})$. We investigate the case where network activity flows toward a low-dimensional manifold $\mathcal{M}$, specifically a unit hypersphere, consistent with the orthogonal transformations derived in our algebraic framework. To determine necessary conditions for a hypersphere attractor, we consider the Lyapunov energy function:

$$E(t) = \frac{1}{2}(\|\mathbf{z}(t)\|^2 - 1)^2, \tag{10}$$

assuming a unit radius ($R = 1$) for simplicity. This energy quantifies the deviation of the state from the manifold surface. The time evolution of the energy along a trajectory in the neural state space is determined via the chain rule. Letting $u = \|\mathbf{z}\|^2 - 1$ denote the deviation from the manifold, we have:

$$\dot{E} \equiv \frac{dE}{dt} = \frac{dE}{du}\frac{du}{dt} = u\dot{u}.$$

The rate of change of the squared norm is derived using the product rule on the inner product $\mathbf{z}^T\mathbf{z}$:

$$\frac{d}{dt}(\|\mathbf{z}\|^2) = \frac{d}{dt}(\mathbf{z}^T\mathbf{z}) = \dot{\mathbf{z}}^T\mathbf{z} + \mathbf{z}^T\dot{\mathbf{z}} = 2\mathbf{z}^T\dot{\mathbf{z}},$$

where the last step follows from the symmetry of the Euclidean inner product ($\mathbf{a}^T\mathbf{b} = \mathbf{b}^T\mathbf{a}$). Combining these terms yields:

$$\dot{E} = (\|\mathbf{z}\|^2 - 1)2\mathbf{z}^T\dot{\mathbf{z}}.$$

Substituting the effective network dynamics from Eq. (9), the energy derivative becomes:

$$\tau\dot{E} = (\|\mathbf{z}\|^2 - 1)2\mathbf{z}^T W[\mathbf{v}]\sigma(\mathbf{z}).$$

This form of the energy derivative implies a specific condition for convergence. If the effective interaction satisfies

$$W[\mathbf{v}]\sigma(\mathbf{z}) = -(\|\mathbf{z}\|^2 - 1)M(\mathbf{v})\mathbf{z},$$

where $M$ is an input-dependent matrix chosen such that the symmetric form is positive definite, then the energy derivative becomes:

$$\tau\dot{E} = -2(\|\mathbf{z}\|^2 - 1)^2\mathbf{z}^T M(\mathbf{v})\mathbf{z} \le 0.$$

Since the quadratic form $\mathbf{z}^T M\mathbf{z}$ is positive, the energy is strictly non-increasing, driving any non-zero neural state toward the hypersphere ($\dot{E} = 0$ only when $\|\mathbf{z}\| = 1$). A simple sufficient condition is for the symmetric part of $M$ to have positive eigenvalues.

We observe that this attractor network admits a fundamental symmetry: the evolution of the energy is invariant under the transformation

$$M(\mathbf{v}) \to M(\mathbf{v}) + \hat{U}(\mathbf{v}),$$

provided $\hat{U}$ is skew-symmetric ($\hat{U}^T = -\hat{U}$). This invariance holds because the quadratic form of any skew-symmetric matrix vanishes identically ($\mathbf{z}^T\hat{U}\mathbf{z} = 0$), meaning $\hat{U}$ does not contribute to the energy derivative. By choosing $\hat{U}(\mathbf{v}) = (\|\mathbf{z}\|^2 - 1)^{-1}U(\mathbf{v})$ and setting $\sigma(\mathbf{z}) = (\|\mathbf{z}\|^2 - 1)\mathbf{z}$, the general hypersphere attractor admits the effective dynamics:

$$\tau\dot{\mathbf{z}} = U(\mathbf{v})\mathbf{z} + M\sigma(\mathbf{z}). \tag{11}$$

This decomposition reveals two distinct functional components: the non-linear term (scaled by $M$) enforces the attractor dynamics normal to the manifold, while the linear term (scaled by $U$) drives

transport along the manifold. Whenever the state is on the manifold, $\mathbf{z}^* \in \mathcal{M}$, the attractor contribution vanishes ($\sigma(\mathbf{z}^*) = 0$). The on-manifold dynamics then reduce to pure transport:

$$\tau\dot{\mathbf{z}}^*(t) = U(\mathbf{v}(t))\mathbf{z}^*(t).$$

For constant velocity inputs, this linear system has the exact solution:

$$\mathbf{z}^*(t) = e^{\frac{t}{\tau}U(\mathbf{v})}\mathbf{z}^*(0).$$

Thus, the exponential map exactly describes the trajectory of the neural state on the attractor manifold. This framework can be realized by a simple recurrent neural network where $U$ is a gain-modulated skew-symmetric matrix (encoding path integration) and $M$ is a positive definite matrix (enforcing stability).

To validate this derivation, we simulated the dynamics of Eq. (11) using Euler integration. We constructed the connectivity matrices as $U = A - A^T$ and $M = B^T B$, where $A, B \in \mathbb{R}^{N \times N}$ were sampled from a normal distribution. These choices ensure that $U$ is skew-symmetric and $M$ is positive definite. We simulated 100 trajectories initialized randomly within a hypercube of side length 0.1, using a time step $dt = 5 \cdot 10^{-5}$ and total duration $T = 0.03$ (for example, seconds, but units are arbitrary). For each trajectory, we computed the Lyapunov energy defined in Eq. (10). Points satisfying the condition $E \leq \varepsilon = 10^{-4}$ were classified as "close points,", representing states effectively on the manifold. Subsequently, we computed the deviation between the evolving network state and the theoretical exponential map trajectory initiated from the first identified close point. The results are shown in Fig. 6 and demonstrate three key properties. First, the Lyapunov decays to zero for all trajectories, indicating that the manifold is attractive and that our construction effectively realizes an attractor network (Fig. 6a). Second, the network state on the manifold is accurately described by the exponential map; the Euclidean distance between the simulated state and the theoretical exponential evolution approaches zero after convergence (Fig. 6b). Finally, to demonstrate that the network state evolves on the manifold, we computed the deviation between the network state and the first close point. As shown in Fig. 6c, this deviation increases after the close point is reached, indicating that the network moves away from $\mathbf{z}^*$ over time, in a manner consistent with the dynamics captured by the exponential map.

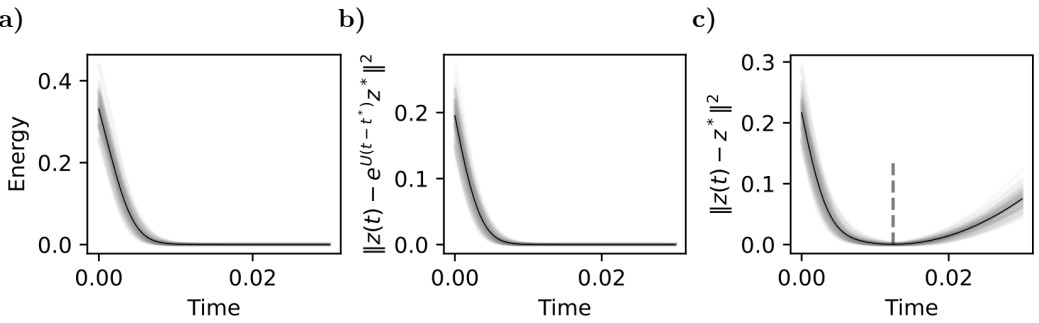

Figure 6: **Exponential maps describe the on-manifold dynamics of a recurrent attractor network.** **a)** Time evolution of the Lyapunov energy of the hypersphere attractor network for 100 trajectories randomly initialized in a hypercube. The gray lines represent individual trajectories and the dark line represents the average. **b)** Squared distance between the network state $\mathbf{z}$ and the theoretical exponential map evolution of the first close point on the attractor ($\mathbf{z}^*$), occurring at $t = t^*$. **c)** Squared distance between network state $\mathbf{z}$ and the static first close point on the attractor ($\mathbf{z}^*$), showing divergence due to transport. The dashed line indicates the average time to convergence $\langle t^* \rangle$.

## H.2 CONNECTION TO THE PATH-INTEGRATING EXPONENTIAL MAP

As we saw in the previous section, we can decompose the weight matrix into a baseline component that sustains the attractor bump and a skew symmetric component that determines the on-manifold dynamics of the attractor network. To bridge this general result with the specific exponential map structure proposed in the main text, we adopt the mechanism of velocity-driven updating standard

in continuous attractor models (Burak & Fiete, 2009; McNaughton et al., 2006). Specifically, we define the transport operator $U$ to be linearly modulated by the self-motion signal $\mathbf{v}(t)$.

By defining $U(\mathbf{v}(t)) = v_x(t)U_x + v_y(t)U_y$, the linearized dynamics become:

$$\tau\dot{\mathbf{z}} = (v_x U_x + v_y U_y)\mathbf{z}$$

The solution to this time-varying matrix differential equation involves the time-ordered exponential ($\mathcal{T}$):

$$\mathbf{z}(t) = \mathcal{T}\left\{\exp\left(\int_0^t \frac{1}{\tau}(v_x(\xi)U_x + v_y(\xi)U_y)d\xi\right)\right\}\mathbf{z}(0).$$

Mathematically, the time-ordered exponential expands into the infinite series known as the Magnus expansion (Blanes et al., 2009). This series includes integral terms involving nested commutators of the operators at different time points, such as, $[U(\mathbf{v}(\xi_1)), U(\mathbf{v}(\xi_2))]$. Expanding this term reveals that it is proportional to the commutator of the basis matrices, $[U_x, U_y]$.

For the system to perform exact path integration, the resulting population vector $\mathbf{p}(x, y)$ must depend solely on the net accumulated displacement, independent of the specific velocity history $\mathbf{v}(t)$ or trajectory taken. This path-independence requires the time-ordered exponential to reduce to a standard matrix exponential of the integrated inputs:

$$\mathbf{z}(t) = \exp\left(\int_0^t \frac{1}{\tau}(v_x(\xi)U_x + v_y(\xi)U_y)d\xi\right)\mathbf{z}(0).$$

This simplification occurs if the commutator terms in the Magnus expansion vanish. Consequently, exact path integration imposes the algebraic constraint that the effective skew-symmetric matrices must commute: $[U_x, U_y] = 0$.

Under this condition, we can integrate the velocity inputs directly. Identifying the spatial coordinates as $x(t) = \int_0^t v_x(\xi)d\xi$ and $y(t) = \int_0^t v_y(\xi)d\xi$, we recover the exponential map model:

$$\mathbf{z}(t) = e^{xG_x + yG_y}\mathbf{z}(0),$$

where the generator matrices are identified as $G_x = \frac{1}{\tau}U_x$ and $G_y = \frac{1}{\tau}U_y$.

This derivation provides a rigorous physical interpretation of the abstract generators: they correspond to the velocity-modulated synaptic connectivity $U$ scaled by the inverse effective time constant $1/\tau$. Furthermore, it demonstrates that the algebraic condition for path independence (commuting generators) derived in the main text is dynamically equivalent to the requirement for a neural circuit to perform exact temporal integration without trajectory-dependent errors.

### H.3 MEMBRANE DYNAMICS AND GRID SCALE HIERARCHIES

The identification of the generator matrices as $G = \frac{1}{\tau}U$ in the previous section implies an intrinsic coupling between the network's time constant $\tau$ and the spatial scale of the resulting representation. To derive this relationship, consider the path-integrating solution on the manifold:

$$\mathbf{z}(x, y) = e^{\frac{1}{\tau}(xU_x + yU_y)}\mathbf{z}_0.$$

As established in Appendix C, the activity of any unit in this representation decomposes into a superposition of plane waves. For a metric-preserving representation, for example, $M = 3$, these waves are generated by rotations in 2D subspaces. The phase angle $\Phi_i$ for the $i$-th subspace is given by:

$$\Phi_i(x, y) = \frac{1}{\tau}(x\lambda_{i,x} + y\lambda_{i,y}),$$

where $\lambda_{i,\cdot}$ are the imaginary parts of the eigenvalues of the connectivity matrices $U$. Converting to polar coordinates with spatial displacement $r$ and head direction $\theta$, the argument becomes:

$$\Phi_i(r, \theta) = \frac{1}{\tau}rk_i\cos(\theta - \phi_i),$$

where $k_i$ represents the intrinsic frequency (magnitude of the eigenvalue) and $\phi_i$ the orientation of the wave. The pattern repeats at integer multiples of $2\pi$. Considering motion along the wave's

propagation direction ($\theta = \phi_i$), the fundamental spatial period $\Lambda_i$ is determined by the condition $\Phi_i(\Lambda_i, \phi_i) = 2\pi$. Solving for $\Lambda_i$ yields:

$$\Lambda_i = \frac{2\pi\tau}{k_i}.$$

This result demonstrates that the period of the constituent plane waves scales linearly with the network's time constant $\tau$.

For a hexagonal grid cell representation ($M = 3$), the global grid pattern is constructed from the interference of three such plane waves. Since the grid spacing corresponds to the distance between pattern repetitions, it is geometrically constrained to be a fixed multiple of the constituent wave periods. Consequently, the grid spacing $\lambda_{\text{grid}}$ must also scale linearly with the time constant:

$$\lambda_{\text{grid}} \propto \tau.$$

This algebraic derivation mirrors the known topographic organization of the medial entorhinal cortex (MEC). Experimental evidence confirms that the membrane time constants of MEC stellate cells increase along the dorsal-ventral axis (Giocomo et al., 2007; Giocomo & Hasselmo, 2008), correlating precisely with the expansion of the grid scale. Furthermore, genetic perturbations that increase the integrative time constant (such as HCN1 channel knockouts) cause a corresponding expansion in grid spacing (Giocomo et al., 2011). Our framework thus provides a normative mathematical explanation for this phenomenon: the spatial scale of the neural map is physically grounded in the integration speed of the underlying neural substrate.

### H.4 EMERGENCE OF SKEW-SYMMETRIC CONNECTIVITY VIA LOCAL LEARNING

We investigate a biologically plausible local learning rule capable of driving the weight matrix $W$ toward skew-symmetry. Motivated by the requirement that the symmetric component of the connectivity must vanish to preserve the manifold energy, we propose the update rule:

$$\Delta W_{ij} = \eta \left( r_i(t) \, r_j(t+\tau) - r_j(t) \, r_i(t+\tau) \right) - \alpha W_{ij},$$

where $r_i$ and $r_j$ denote pre- and postsynaptic activities, $\tau$ represents a small temporal lag, $\eta$ is the learning rate, and $\alpha$ is a local decay term.

Decomposing the weight matrix into symmetric ($S = W + W^T$) and antisymmetric ($A = W - W^T$) components reveals distinct evolutionary dynamics. Since the Hebbian term $H_{ij} = r_i(t)r_j(t+\tau) - r_j(t)r_i(t+\tau)$ is inherently antisymmetric ($H_{ij} = -H_{ji}$), it contributes zero to the update of the symmetric component. Consequently, the dynamics of $S$ are governed solely by the decay term:

$$\Delta S = -\alpha S.$$

This ensures that any initial symmetric connectivity decays asymptotically to zero. Conversely, the antisymmetric component is reinforced by the Hebbian term:

$$\Delta A = 2\eta \left( r_i(t) \, r_j(t+\tau) - r_j(t) \, r_i(t+\tau) \right) - \alpha A.$$

Thus, $A$ is driven by the time-lagged anti-correlated activity patterns while being stabilized by the decay $\alpha$. In the steady state, the antisymmetric part converges to:

$$A^* = \frac{2}{\alpha} \left\langle r_i(t) \, r_j(t+\tau) - r_j(t) \, r_i(t+\tau) \right\rangle_t,$$

where the brackets denote a time average. This implies that the learned skew-symmetric connectivity is determined by the statistics of the time-lagged antisymmetric correlations in the network activity. By controlling these correlations, the network can, in principle, learn specific generator structures.

To validate this mechanism, we simulated the training of a $32 \times 32$ weight matrix. We initialized the weights and the neural activity rates from a normal distribution, with parameters set to $\alpha = 0.001$ and $\eta = 0.01$. The evolution of the weights was tracked over 5000 iterations. The results, shown in Fig. 7, demonstrate that the magnitude of the symmetric component decays to zero, while the skew-symmetric component persists. This confirms that the proposed rule effectively filters out symmetric connectivity, allowing skew-symmetric generators to emerge naturally from local plasticity rules.

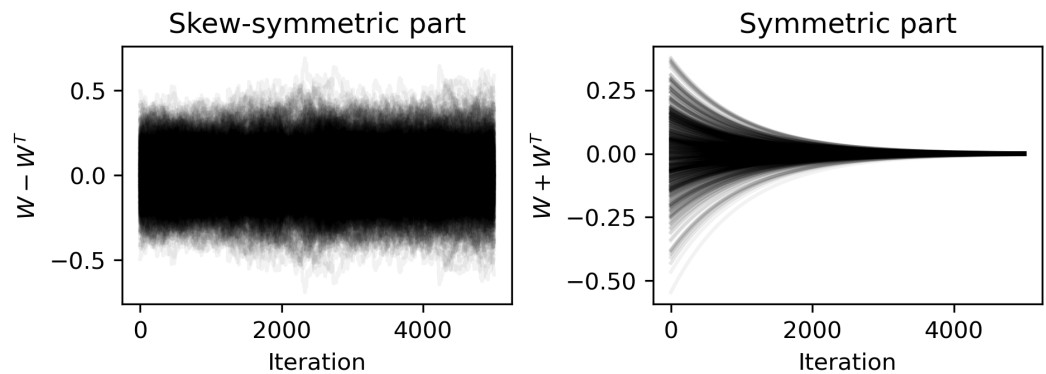

Figure 7: **Emergence of skew-symmetric connectivity via local learning.** Evolution of weight matrix components under the anti-symmetric Hebbian learning rule. The symmetric component (decaying curve) vanishes over time, while the skew-symmetric component (fluctuating curve) persists, driven by the correlation structure of the inputs.

## I LEARNING EXPONENTIAL MAPS FROM DATA

To validate the practical applicability of our theoretical framework, we demonstrate that the exponential map model can learn to reproduce and generalize experimental neural data. Using publicly available recordings of grid cells from rats navigating an open field (Gardner et al., 2022), we trained the model to capture the generative structure of the spatial code.

We formulated a one-step predictive model where the predicted firing rate $\hat{\mathbf{g}}$ at a target location is generated from the rate at a source location via the matrix exponential. For a spatial displacement $(\Delta x, \Delta y)$ corresponding to small integer index steps, the update rule is:

$$\hat{\mathbf{g}}[x + \Delta x, y + \Delta y] = e^{\Delta x G_x + \Delta y G_y}\hat{\mathbf{g}}[x, y],$$

where $G_x$ and $G_y$ are learnable generator matrices. We optimized the model using stochastic gradient descent to minimize a composite loss function:

$$\mathcal{L} = \frac{1}{K}\sum_k^K \|\mathbf{g}_{target} - \hat{\mathbf{g}}_{pred}\|^2 + \frac{1}{N^2}\sum_{i,j}^{N^2}((G_x G_y)_{ij} - (G_y G_x)_{ij})^2.$$

The first term represents the reconstruction error between the predicted activity and the experimental rate map $\mathbf{g}$ (averaged over a batch size $K$). The second term is a commutation penalty, which enforces the algebraic constraint $[G_x, G_y] \approx 0$. This constraint is critical for ensuring path independence and allows the model to learn consistent spatial maps without requiring computationally expensive multi-step training sequences.

For the training protocol, generators $G_x$ and $G_y$ were initialized as random uniform matrices scaled by $1/\sqrt{N}$, where $N$ is the number of simulated neurons. Optimization was performed using the Adam optimizer (Kingma & Ba, 2017) with a learning rate of $0.001$ for $20,000$ iterations. Training samples were generated by selecting random starting locations within the experimental rate maps and taking single steps of up to two pixels; boundary effects were mitigated by reflecting steps that exceeded the rate map limits.

To evaluate the model's generative capacity, we reconstructed global rate maps by path integrating from the center of the arena (origin) to a dense grid of position coordinates $(x, y)$:

$$\mathbf{p}(x, y) = e^{x G_x + y G_y}\mathbf{p}(0, 0).$$

To test extrapolation, we extended this coordinate grid to cover an area with twice the side length of the original experimental enclosure.

The results, shown in Fig. 8, demonstrate that the exponential map not only reproduces the training data but successfully generalizes the grid pattern well beyond the boundaries of the original recording enclosure (indicated by the black square). The loss history (Fig. 9) confirms that the model simultaneously minimizes reconstruction error and enforces commutativity. This indicates that the commutation constraint enables the model to extract the robust, intrinsic algebraic structure of the spatial representation directly from noisy biological data. We note, however, that the learned rate maps exhibit more uniform peak firing rates than their biological counterparts. We hypothesize that this uniformity may be an emergent consequence of the commutation penalty and the strict path invariance it enforces. While this idealization abstracts away biological heterogeneity, which may stem from noise or conjunctive inputs not modeled here, it allows the framework to robustly capture the intrinsic spatial phase and periodicity of the grid pattern.

These results suggest several promising avenues for future inquiry. First, future work should compare the remapping dynamics of metric-preserving model units (Fig. 3) with biological recordings to determine if they exhibit the coherent remapping characteristic of entorhinal and hippocampal ensembles (Fyhn et al., 2007). Second, fitting exponential map models to other cell types, such as place cells, offers a pathway to deriving interpretable models of their underlying dynamics. Specifically, it remains to be determined whether the exponential map can account for the heterogeneous, apparently stochastic spatial arrangement of biological place fields. A compelling validation would involve fitting the model to rate maps recorded in restricted enclosures and generating predictive extrapolations for larger environments. Comparing these predictions against experimental data from expanded arenas would rigorously test the model's ability to capture the intrinsic generative structure of the spatial code.

## J   BINDING AND BUNDLING AS A BASIS FOR MULTIPLE-MAP, REWARD-ORIENTED NAVIGATION

While our primary focus has been on the geometric and computational properties of spatial representations, navigation involves more than localization. In this section, we demonstrate that the exponential map framework supports reward-based navigation in an interpretable manner by drawing on principles from Hyperdimensional Computing (HDC) (Kanerva, 2009).

HDC encodes information using distributed representations based on high-dimensional vectors, commonly termed hypervectors. A core property of such high-dimensional spaces is the "concentration of measure," which ensures that randomly sampled vectors become nearly orthogonal with high probability with increasing vector dimension. HDC systems exploit this phenomenon via two primary operations. The first is *bundling*, or superposition, which aggregates vectors to form a composite representation that remains similar to its inputs. The second is *binding*, an operation that combines vectors to produce a result that is dissimilar, and effectively orthogonal, to its constituents. We show that the exponential map naturally implements these operations, thereby tying together spatial representation, memory, and goal-oriented behavior.

### J.1   BUNDLING: CONSTRUCTING REWARD MAPS

Assuming a similarity-preserving representation, such as the Gaussian similarity derived in Section 2.5, we can implement a simple memory mechanism by "bundling" the population vectors of salient locations. Let $\{\mathbf{r}_i\}_{i=1}^{N_R}$ be a set of locations associated with a reward. A composite memory vector $\mathbf{p}_R$ is formed by summing the representations:

$$\mathbf{p}_R = \frac{1}{N_R} \sum_i^{N_R} \mathbf{p}(\mathbf{r}_i).$$

Querying the current location $\mathbf{p}(\mathbf{r})$ against this memory yields a similarity score $C_R$:

$$C_R(\mathbf{r}) = \mathbf{p}(\mathbf{r})^T \mathbf{p}_R = \frac{1}{N_R} \sum_i^{N_R} \mathbf{p}(\mathbf{r})^T \mathbf{p}(\mathbf{r}_i).$$

Since the pairwise similarity approximates a Gaussian, the resulting score $C_R(\mathbf{r}) \approx \frac{1}{N_R} \sum_i e^{-\|\mathbf{r}-\mathbf{r}_i\|^2/\sigma^2}$ acts as a smooth reward density map (a mixture of Gaussians). This allows an

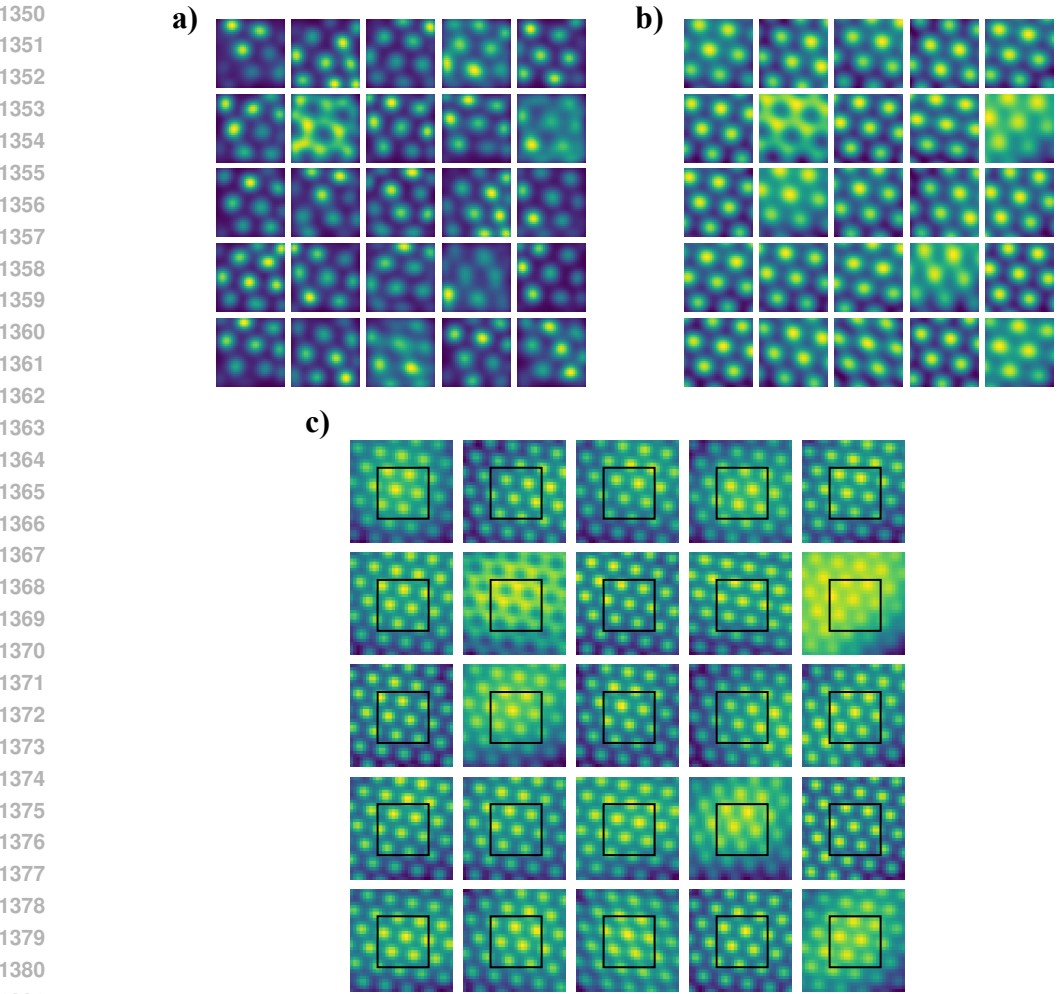

Figure 8: **Generative modeling and extrapolation of experimental grid cells. a)** Experimental grid cell rate maps used for training. **b)** Synthetic rate maps generated by the exponential map model within the training domain. **c)** Extrapolated rate maps in the extended environment. The model captures the grid structure within the training domain and successfully extrapolates the periodic pattern beyond the boundaries (indicated by the black outline). Rate maps are matched cell-to-cell across rows.

agent to navigate to rewards by simply following the gradient of the similarity surface:

$$\mathbf{r}_{t+1} \leftarrow \mathbf{r}_t + \eta \nabla_{\mathbf{r}} C_R,$$

where $\eta$ is a step size. Biologically, this gradient ascent could be approximated by sampling local steps and moving in the direction of increasing similarity.

### J.2 BINDING: CONTEXT-DEPENDENT MAP RETRIEVAL

While bundling creates a single reward map, complex navigation requires storing distinct maps for different contexts, for instance distinguishing between a "Food" context and a "Home" context. The standard HDC "binding" operation, often implemented via element-wise multiplication, orthogonalizes vectors. In our framework, context-dependent remapping functions as an intrinsic binding operation.

Recall that the context generator $G_s$ produces orthogonal transformations. If the remapping is sufficiently strong, implying dissimilar contexts, the representations become nearly orthogonal. We can thus form a general, context-dependent memory vector $\mathbf{p}_{R,S}$ that bundles spatial locations across

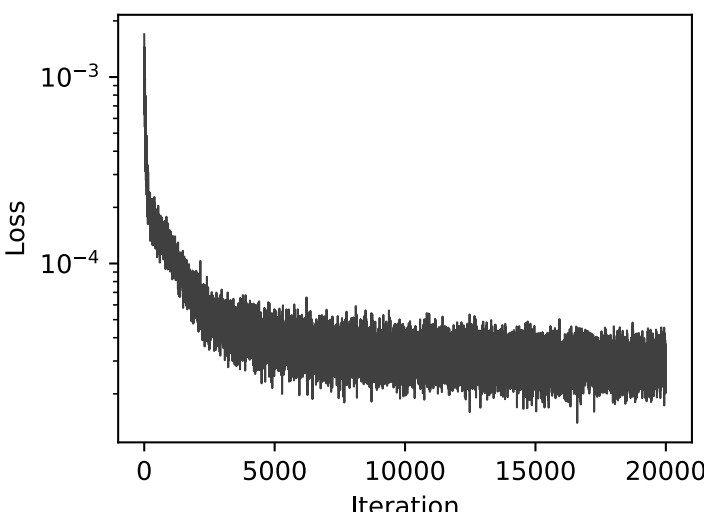

Figure 9: **Optimization and commutation dynamics.** Evolution of the total loss function during the optimization of the exponential map model on experimental grid cell data.

different contexts, such as rooms $s_1$ and $s_2$:

$$\mathbf{p}_{R,S} = \sum_{i=1}^{N_1} \mathbf{p}(\mathbf{r}_i, s_1) + \sum_{j=1}^{N_2} \mathbf{p}(\mathbf{r}_j, s_2) \tag{12}$$

$$= e^{s_1 G_s} \sum_i \mathbf{p}(\mathbf{r}_i) + e^{s_2 G_s} \sum_j \mathbf{p}(\mathbf{r}_j). \tag{13}$$

When we query this composite memory with a current state in context $s_1$, the orthogonality of the remapping filters out the interference from context $s_2$:

$$C_{R,S}(\mathbf{r}, s_1) = \mathbf{p}(\mathbf{r}, s_1)^T \mathbf{p}_{R,S}$$

$$= \mathbf{p}(\mathbf{r}, s_1)^T \sum_i \mathbf{p}(\mathbf{r}_i, s_1) + \mathbf{p}(\mathbf{r}, s_1)^T \sum_j \mathbf{p}(\mathbf{r}_j, s_2)$$

$$\approx C_{R1}(\mathbf{r}) + 0.$$

Here, the cross-context term vanishes because the relative context shift $\Delta s = s_2 - s_1$ decorrelates the vectors (as shown in Fig. 3). This mechanism allows for the superposition of multiple, distinct cognitive maps within a single neural population, enabling context-specific retrieval without crosstalk.

Figure 10 illustrates this capability. We constructed a memory vector summing three target locations across two contexts ("Home" and "Food"). Querying with the appropriate context signal retrieves the correct spatial map, guiding gradient-based navigation to the relevant targets. This demonstrates that the exponential map framework naturally supports flexible, goal-oriented navigation through the algebraic composition of bundling and binding operations.

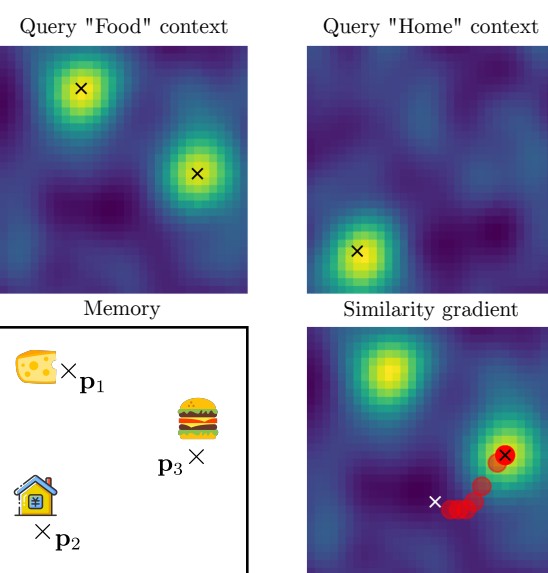

Figure 10: **Multi-map, reward-oriented navigation using exponential maps. Bottom left:** A composite memory vector stores the locations of three objects (Cheese, Hamburger, Home) belonging to two distinct contexts (Food and Home). **Top left:** Querying the memory with the "Food" context retrieves a similarity map highlighting the two food items. **Top right:** Querying with the "Home" context retrieves the home location. **Bottom right:** Gradient ascent on the retrieved "Food" similarity map generates a trajectory towards the rewards. The similarity function was constructed to be approximately Gaussian ($N = 300$).

