# OpenReview forum: "EXPONENTIAL MAP MODELS AS AN INTERPRETABLE FRAMEWORK FOR GENERATING NEURAL SPATIAL REPRESENTATIONS"
_ICLR.cc/2026/Conference — Submitted to ICLR 2026_

### Official Review · Reviewer_zmDx · 2025-10-23

**Soundness:** 4
**Presentation:** 3
**Contribution:** 4
**Rating:** 10
**Confidence:** 5

**Summary:**

This paper introduces a first-principles framework for generating neural spatial representations using exponential map models based on generator matrices. Rather than relying on deep learning's "black-box" approaches, the authors derive exact algebraic conditions for key properties of neural spatial maps: (1) commuting generators ensure path-independent representations for reliable path integration, (2) skew-symmetric generators produce translationally invariant similarity structures ideal for egocentric navigation, and (3) generator eigenvalues forming roots of unity preserve the flat metric of space. The framework generates diverse biologically plausible spatial tuning including place cells, grid cells, and context-dependent remapping.

**Strengths:**

1. Novel theoretical contribution: The use of generator matrices with the matrix exponential to construct spatial representations is elegant and, to my knowledge, novel in this explicit form. The mathematical framework is transparent and interpretable, addressing a key limitation of deep learning approaches.

2. Rigorous mathematical development: The derivations are exceptionally clear and well-structured. The authors systematically build from basic exponential maps to derive exact algebraic conditions for path invariance, translational invariance, and metric preservation. The appendices provide thorough derivations for readers seeking additional detail.

3. Excellent writing quality: The paper is very well written, with clear explanations that balance mathematical rigor with intuitive understanding. Figure 1 provides an excellent conceptual overview.

4. Biological relevance: The framework generates diverse neural tuning curves (grid cells with different symmetries, place-like cells) and can model remapping behavior. The connection to modular organization of grid cells and the Bessel function analysis (Appendix F) showing spacing ratios similar to experimental observations (√2) is particularly compelling.

5. Unification: The framework elegantly unifies multiple spatial representations under a single mathematical structure, showing how different tuning properties emerge from different choices of generator eigenvalues and similarity functions.

6. Transparency over black-box models: Unlike RNN-based approaches, this framework allows exact specification of which mathematical conditions produce which navigational properties, making it highly interpretable.

**Weaknesses:**

## Major Issues

1. Limited contextualization with prior work:
  * The introduction cites only Ginosar et al. (2023) regarding grid cells providing a metric for space, but there is substantial earlier literature on this topic that should be acknowledged.
  * Whittington et al. (2020) is characterized as a "black-box" model, but it actually makes explicit theoretical claims about spatial representations based on prior work by others. The distinction between truly black-box models (Banino et al., Cueva & Wei) and more theory-driven approaches should be clearer.
  * The relationship between this work and previous approaches could be more explicitly stated in the introduction, though it becomes clear later in the paper.

2. Missing connections to transition/successor representation literature:
  * Lines 226-228 claim that similarity translational invariance has not been shown before "to the best of our knowledge." However, related concepts appear in prior work on transition coding and successor representations in grid cells (Stachenfeld et al. 2017, Waniek 2018 & 2020, possibly Rebecca et al. 2025 on spatial periodicity). These should be checked and cited if relevant.
  * Section 2.4 states the metric preservation goal "as proposed by (Gao et al., Xu et al.)" but similar computational properties for grid cells were also motivated in the transition coding literature that I just mentioned

3. Incomplete biological implementation discussion:
  * Around the end of Section 2.2, it would strengthen the paper to briefly discuss what the mathematical findings mean in terms of potential neural implementation of G_x and G_y. How might biological circuits realize these generator matrices?

4. Limited discussion of learning mechanisms:
  * Section 4 (Limitations) acknowledges this gap, but the paper would be stronger with at least brief speculation about how biologically plausible learning rules might converge to these solutions. The authors mention this as future work, but some initial thoughts would be valuable.


## Minor Issues

1. Awkward phrasing: Line ~111: "Equation (1) does what we intended it to;" is oddly casual and should be reworded more formally.
2. Paragraph flow issues:
  * The paragraph after Equation (3) is difficult to unpack. Reordering the sentences could improve clarity.
  * The final paragraph of Section 2.3 is somewhat unclear and could be revised for better flow.
3. Digression: Lines 228-234 contain an interesting but substantial digression about batch/layer normalization that disrupts the main narrative. Consider moving this to a footnote or brief remark.
4. Exponential family connection: Line 375 discusses the Gaussian as a special case. This could potentially be extended to the exponential family with linear sufficient statistics. Prior work by Pouget (Beck et al. 2007) suggests neural responses should fall into this category, which might be worth discussing.
5. LLM usage statement location: The ICLR 2026 guidelines require the LLM usage statement in the main body, not just the appendix. Please move the statement from Appendix A to the main text.
6. Notation clarity: While generally clear, ensuring consistent notation throughout (especially regarding conjugate eigenvalues and the block structure) would help readers following the mathematical derivations.

## Minor Corrections
1. Check citation formatting throughout
2. Ensure all figures are referenced in order in the text
3. Consider adding a table summarizing the algebraic conditions and their corresponding properties for easy reference

**Questions:**

1. Can you clarify the relationship between your similarity translational invariance result and prior work on successor representations and transition systems in spatial coding?
2. How do you envision the generator matrices G_x and G_y being implemented or learned in biological neural circuits?
3. In the remapping model (Section 2.5), you use a single scalar context signal s. How would the framework extend to high-dimensional context representations (e.g., visual scenes)?
4. The choice of orthogonal matrix R remains free in your framework. Beyond energy constraints or non-negativity, are there other biologically motivated constraints that might determine R?
5. Could you expand on the connection to the exponential family (beyond Gaussians) for the similarity preservation framework?

---

> ### Author Response · Authors · 2025-11-25
> **Author Response**
>
> We are grateful for the "strong accept" and the reviewer's recognition of the novelty and rigor of our framework. We have incorporated your suggestions to further strengthen the manuscript.
>
> ### 1.  Biological implementation (Re: Weakness 3 & Question 2)
>
> We have addressed this extensively in the new **Appendix H** (see **Global Response**). We now explicitly discuss how biological circuits implement the generator matrices. Specifically, we show that we recover the exponential map model as the exact description of the on-manifold dynamics of a CANN, implementable using a simple RNN. Using the CANN formulation also links grid spacing and time constants, consistent with experimental observations.
>
> ### 2. Connection to other theoretical models of spatial coding (Re: Weakness 1 & Question 1)
>
> We have added citations to Stachenfeld et al. (2017), Waniek (2018, 2020), Whittington et al. (2020),  and Rebecca RG et al. (2025). In the revised **Introduction**, **Section 2.3** discussion, we position our work as complementary: while SR relies on the statistics of transitions (eigenvectors of $T$), our work derives similar structures from the algebraic constraints required for path integration (a geometric view). Our condition for translational invariance ($G^T = -G$) ensures orthogonal transformations, which parallels the spectral properties found in translation-invariant transition systems. Both approaches arrive at grid-like solutions, but our framework provides explicit algebraic conditions that do not depend on sampling a transition matrix.
>
> ### 3. High-dimensional contexts (Re: Question 3)
>
> You asked about extending the scalar context $s$ to high dimensions. The algebra naturally supports this extension: the context generator $sG_s$ becomes a vector product $\mathbf{s} \cdot \mathbf{G_s}$, where $\mathbf{G_s}$ is a tensor of generators. In the revised **Limitations** section, we discuss how this could model complex inputs like visual scenes, where "similarity preservation" would mean preserving the semantic distances between images. We find this to be a very interesting extension of our work, and hope to explore this direction in the future.
>
> ### 4. Choice of matrix R (Re: Question 4)
>
> We agree that $R$ is currently a free choice (random orthogonal). We have added a discussion noting that biological constraints, such as non-negativity (leading to firing rates) or metabolic sparsity, would likely constrain $R$, forcing the abstract plane waves into the specific lattice-like activity bumps seen in biology. Most notably, Xu. et al. (2025) showed that a hexagonal, grid-like solution (M = 3) is the optimal solution for extrinsic distance preservation—that is, preserving global Euclidean distances, not just the local metric—which provides clear utility for navigation. We have added this point to the limitations section.
>
> ### 5. Minor issues
>
> We have fixed the phrasing of Equation (1), improved the flow in Section 2.3, and moved the LLM disclosure to the main body as per ICLR guidelines.
>
> **Exponential family connection:** We thank the reviewer for highlighting this important connection. We agree that the spatial tuning curves generated by our model are fully compatible with the Probabilistic Population Codes (PPC) framework (Ma et al., 2006; Beck et al., 2008). Specifically, if neural variability follows a distribution in the exponential family (e.g., Poisson-like) with linear sufficient statistics, the Gaussian-like tuning curves derived in Section 2.5 can support optimal Bayesian inference via linear integration. In this context, our algebraically derived representations effectively define the tuning kernel $h(s)$ within the PPC formalism. We will add a remark to the discussion acknowledging this synthesis.

---

### Official Review · Reviewer_XtHm · 2025-10-28

**Soundness:** 4
**Presentation:** 4
**Contribution:** 3
**Rating:** 6
**Confidence:** 4

**Summary:**

This submission develops a simple and interpretable framework for generating and understanding spatial representations that are found in the hippocampal formation. The authors show how different properties that are presumably important for spatial representations shapes the kinds of representations that emerge in their framework, and they demonstrate that these are aligned with experimentally observed functional classes, such as grid cells and place cells.

**Strengths:**

1. The paper was very well written and easy to follow.

2. Figure 1 was very nicely done and helped make the work more clear.

3. The transparent framework is a welcomed compliment to the RNN models that have become very popular.

4. The found sqrt(2) ratio of Bessel function 0s is really interesting and fits nicely with the experimentally found ratio in grid module spacing.

5. The fact that the same framework can find both grid and place cells is nice, and the interpretable difference between the conditions that give rise to them is cool.

**Weaknesses:**

1. The only major weakness I think is just that the framework, while interesting and interpretable, lacks a really clear punch. I hate getting this comment in my own work, but I was left feeling a little bit like "what did I learn from this?" This is especially the case since some of the different elements in this work have been explore previously (definitely not in as coherent or complete a framework as this though).
     My one thought for addressing this is to think a little bit more about the place cell results. If you make the environment really big, and add the same requirements, do you see anything resembling the multi-place fields that are seen in big spaces? Alternatively, the place cell responses aren't perfectly "place cell"-like. Is this being driven by the need to have the correlation decrease from the center of the environment? Are experimentally recorded place cells that do not have as nice place fields actually more informative about distance from center, when taken as a population?
     All of this is just to say that using the model to make some prediction that could be compared to experiments would make it more impactful.

Minor points:

1. I have always been a little confused by the conformal isometry argument. Grid cells from the same module, because they are viewed to be identical (up to phase), can't form a large-scale conformal isometry, right? They can only do so over a small region (up to I guess half the grid spacing). I think making it clear that this requirement is assumed to hold over small $\Delta_x$ and $\Delta_y$ would be helpful in making this more clear.

2. I didn't really feel like Figure 1f was so helpful in understanding the remapping. Are each of those the same population vector, but for different sensory inputs?

3. I thought it was interesting that the higher order M representations (Fig. 2) lead to responses that look almost honeycomb like. This was seen recently in the RNNs trained by Redman et al. (2024) NeurIPS on dual agent path integration. Maybe your framework can provide some insight on why this could occur?

4. Very minor, but Ginosar et al. (2023) is cited as evidence that grid cells provide a metric of space. But that paper largely argues that it is not a metric (at least, not in the global sense). So maybe it'd be more representative to pick a different paper to reference.

**Questions:**

1. Are there any other experimental findings that your framework sheds light on, beyond just the emergence of place and grid cells?

---

> ### Author Response · Authors · 2025-11-25
> **Author Response**
>
> We thank the reviewer for the excellent score and the encouraging comments on the clarity of our work.
>
> ### 1. The "Punch" (Re: Major Weakness)
>
> To address the question of the work's conceptual significance, we have added **Appendix H** (see **Global Response**). The conceptual "punch" is now centered on bridging the gap between algebraic theory, biological mechanism, and functional navigation. First, we demonstrate that the exponential map is not just an abstract model but exactly describes the on-manifold dynamics of a Continuous Attractor Neural Network (CANN). This mechanistic link yields concrete biological predictions, specifically that grid cell spacing scales linearly with the membrane time constant ($\lambda \propto \tau$), a relationship that matches experimental data. Second, we validate the model's predictive power in **Appendix I**, showing that an exponential map fitted to experimental grid cell data allows for accurate extrapolation of the spatial code beyond the training environment. Finally, we demonstrate functional utility in **Appendix J**, showing how the framework enables explicit, multi-map, reward-oriented navigation by exploiting the orthogonalization provided by remapping. Together with our interpretation of grid scales via Bessel zeros, this work presents a unified, transparent, and rigorous framework for understanding the fundamental principles of spatial navigation.
>
> ### 2. Conformal isometry (Re: Minor Point 1)
>
> This is an important clarification; thank you for bringing it up. At its heart, conformal isometry is a local property that measures the warping or stretching of space at a single point. As such, grid cells are capable of encoding a conformal isometry (metric preservation) over a large region, because space is warped equally at every point. However, grid cells cannot preserve extrinsic distances over large regions, meaning physical distances cannot be computed directly from population vectors globally. This limitation was explored by Pettersen et al. (2024) and Xu et al. (2025), who showed that while grid cells are optimal representations, they can only reliably preserve extrinsic distance up to a finite order. We have updated the manuscript to clarify that our metric preservation argument holds locally and requires disambiguation (via multiple modules) for global distance preservation.
>
> ### 3. Figure 1f and remapping (Re: Minor Point 2)
>
> We apologize for the confusion. We have rewritten the caption for Figure 1f. The "sheets" represent the spatial activity map of the same neural population but under different values of a non-spatial context signal $s$. As $s$ varies, the map shifts (remaps). Crucially, similar contexts (neighboring sheets) map to nearby locations in the representational space, preserving the similarity structure of the context variable.
>
> ### 4. Higher-order symmetries and Redman et al. (2024) (Re: Minor Point 3)
>
> We appreciate the reference to Redman et al. (2024). Our framework offers an explicit algebraic explanation for these "honeycomb-like" structures: they emerge naturally when the generator eigenvalues form sets of roots of unity with higher-order symmetries (e.g., $M=4$, as shown in Fig. 2). Since our model constructs these representations via the superposition of plane waves, this suggests that the RNNs in Redman et al. likely converged to similar higher-order root-of-unity solutions to satisfy the geometric constraints of their dual-agent path integration task.
>
> ### 5. Ginosar et al. citation (Re: Minor Point 4)
>
> We have adjusted the citation context and added references to the broader literature.

---

> > ### Comment · Reviewer_XtHm · 2025-11-26
> >
> > I thank the authors for the extensive updates in the rebuttal and their comprehensive answering to all my questions.
> >
> > I think the punch of the paper is now greatly enhanced and I imagine that the developed framework will be of interest to the community.
> >
> > I have one remaining question regarding the results in Fig. 8. There seems to be a little mismatch between the experimental ratemaps and the learned ratemaps, in terms of the variability in grid fields. For instance, a striking feature of the upper leftmost ratemap (bottom) is that not all the fields have the same strength. In contrast, the learned fields all have the same magnitude. Is this due to an explicit assumption of the model that all fields have the same magnitude? I also think the point might be made stronger if 3 subpanels were used (or the figure was split into 2). One with the experimental data, one with the learned grid fields - in the same size environment - and one with the learned grid fields in the larger environment. This would just make it easier to directly compare the fitted ratemaps in the smaller environment with the experimentally recorded ratemaps.

---

> > > ### Author Response · Authors · 2025-12-03
> > > **Author Response**
> > >
> > > We thank the reviewer for their positive assessment and are glad to hear that you think the new results and revisions have greatly enhanced the paper.
> > >
> > > Regarding your specific question on grid field variability, you are correct that experimental rate maps often show variability in peak firing rates, whereas our learned model produces fields with fairly uniform magnitude. We find this "more-uniform-than-data" property interesting. Since we do not explicitly constrain the field shape or amplitudes, we suspect this uniformity might be a consequence of the commutation penalty and the strict path invariance it enforces. Alternatively, the variability in the biological data might stem from the conjunctive encoding of cues not currently included in our model, such as head direction. An interesting next step would be to extend the exponential map with biologically plausible conjunctive inputs to see if this induces more realistic field variability. Thank you for raising this point; it highlights the distinction between our model, which idealizes the amplitude profile to capture the geometric structure, and the biological system, where variability may reflect noise, non-uniform inputs, or specific attractor dynamics. We have added a note to Appendix I to explicitly acknowledge this.
> > >
> > > We also agree that separating the visualization into distinct panels makes the comparison clearer. We have updated Figure 8 (now split into sub-panels A, B, and C) as suggested.

---

### Official Review · Reviewer_xaMZ · 2025-10-30

**Soundness:** 3
**Presentation:** 2
**Contribution:** 2
**Rating:** 2
**Confidence:** 4

**Summary:**

This manuscript developed a mathematical framework for generating neural spatial representations using an exponential map model.  The work builds up recent studies on studying path integration and grid cells by Gao et al (2021), McNamee et al (2021), Xu et al (2022), and also consider the extension to other types of spatial representations. The authors show that under certain conditions, this model can generate spatial representations that bear some similarity to some neuroscience observations,  such as grid cells and place cells.

**Strengths:**

The mathematical framework is elegant and for the most part principled  (although, when it comes to particular modeling settings for explaining the biological observations,  some assumptions need further justification).

By varying components of the model, it can generate neural representations that, in broad strokes, similar to spatial representations observed in the brain, in particular, in the rodent brain.

The limitations of the work were carefully discussed.

**Weaknesses:**

The writing can be improved. While some of the materials are inherently technical, I should there should be better ways to organize and justify the ideas and assumptions.


While the mathematical framework is elegant, it is also quite post-hoc. The explanation power of the framework is questionable. In particular, whether unique or new insights that can be derived from this framework remains unclear, given most of the basic phenomena in neuroscience covered in the paper can also be explain by various alternative models. I think highlighting the unique contribution of the work beyond prior work would be important for the revised version of the paper.

The authors show that by changing M, different tuning patterns can be generated, such as square grid for M=2 and hexagonal grid for M=3. This is interesting. But it is also not surprising, given that these are basically superpositions of plan waves. I think the paper would be made much stronger if the theoretical framework can be used to explain why some network training schemes lead to square grids, while others lead to hexagonal grids or other pattern.

The comparison to the experimental data is vague and could be made more rigorous.

Line 226- 230 stated that the similarity translational invariance has not been explored explicitly in prior work.  I personally find that the similarity translations invariance is not surprising, and was present in many prior studies on neural coding, e.g., neurons with shift-invariant tuning curves that cover a continuous dimension. Place cells on a ring would be an example of this.

**Questions:**

(1) Line 251-253: The authors stated “The translational invariance induced by skew-symmetric generators comes with a non-trivial ad- vantage: The similarity is invariant to a constant, non-spatial shift, similar to remapping behavior (see Section 2.5 for details and Fig. 1f) for an illustration) (Leutgeb et al., 2004; Fyhn et al., 2007). ” Can this be unpacked and can this be directly tested based on experimental data?

(2) Perhaps related to last question:
The authors argued that hippocampal remapping can be explained by this framework. What are the specific predictions regarding remapping that could be empirically tested? At a first look, it seems that the theory predicts a preservation of the similarity structure for spatial across contexts.  Does that mean the similarity structure of each pair of place cells should be preserved?


(3) Can the authors explain what unique insights can be learned from this work?

(4) Can the theoretical framework can be used to explain why some network training schemes lead to square grids, while others training schemes lead to hexagonal grids?

---

> ### Author Response · Authors · 2025-11-25
> **Author Response**
>
> We thank the reviewer for their critique. We have worked to address the concerns regarding the "post-hoc" nature of the model by strengthening the mechanistic derivations.
>
> ### 1. Explanatory power & unique insights (Re: Weakness 2 & Question 3)
>
> We respectfully disagree that the insights are purely post-hoc. As detailed in the **Global Response**, we have added derivations that link our algebraic constraints to specific biological phenomena that standard RNNs do not explain analytically. Our derivation establishes that the algebraic generators $G$ correspond physically to the velocity-modulated synaptic connectivity. This result subsequently suggests that grid spacing is linearly related to the network time constant, providing a mathematical justification for the dorsal-ventral expansion of grid scales (Giocomo et al., 2007). These are normative insights: they explain *why the biology* looks the way it does based on the algebraic necessity of defining a consistent spatial map.
>
> ### 2. Translational invariance (Re: Weakness 5)
>
> You noted that similarity translational invariance is not surprising. We agree that it is a known property of attractor models, and we have revised the manuscript to reflect this. However, the novelty here lies in identifying the exact algebraic condition (skew-symmetry) that guarantees this property for high-dimensional vector representations within an exponential map formalism. This proves that translational invariance implies a specific generator structure, significantly narrowing the search space for valid neural codes. Furthermore, this insight gains significant weight in light of our new attractor network derivation (**Appendix H**), which demonstrates that these algebraic constraints translate directly into the synaptic connectivity (weights) of a Continuous Attractor Neural Network (CANN).
>
> ### 3. Square vs. hexagonal grids (Re: Question 4)
>
> Our theory establishes the "menu" of valid solutions (roots of unity) that allow for metric preservation. Both square ($M=2$) and hexagonal ($M=3$) solutions are algebraically valid. The specific selection of $M=3$ in biology likely arises from downstream constraints, such as packing efficiency or metabolic cost, which act as regularizers on the algebraic solution space. Perhaps most notably, as shown by Pettersen et al. (2024) and Xu et al. (2025), the hexagonal case is the optimal periodic solution for preserving extrinsic distances into the representation. We have clarified these points in the discussion.
>
> ### 4. Remapping predictions (Re: Questions 1 & 2)
>
> Our remapping model (**Section 2.5**) predicts that while relative spatial similarity is preserved within a context, population vectors become orthogonal as the context signal $s$ varies. Consequently, while the spatial relationships between individual place cells may shift, the spatial similarity structure of the population vectors is maintained even as contextual similarity decreases. To elucidate this, we added **Appendix J**, demonstrating that these properties enable the exponential map to support a simple, interpretable form of multi-map, reward-oriented navigation. Furthermore, we propose that future experiments could infer an exponential map model from an animal navigating in a limited context and use it to predict representations in unseen environments. In fact, we demonstrate the feasibility of this approach in **Appendix I**, where we show that an exponential map fitted to experimental grid cell data allows for accurate extrapolation of the spatial code beyond the boundaries of the training environment.

---

### Official Review · Reviewer_En7c · 2025-11-01

**Soundness:** 3
**Presentation:** 2
**Contribution:** 2
**Rating:** 4
**Confidence:** 4

**Summary:**

This paper proposes a transparent and mathematically interpretable framework for constructing neural spatial representations using exponential map models. Instead of relying on deep neural networks or training-based optimization, the authors define spatial encoding through matrix exponentials of generator matrices,Gx Gy, which map physical locations to neural population vectors. The paper identifies several algebraic conditions that guarantee biologically meaningful and geometrically coherent representations, e.g., commutativity, skew-symmetry and roots of unit as eigenvalues.

**Strengths:**

- Conceptual Clarity and Theoretical Rigor

The paper provides a clear derivation of spatial representations from first principles, linking algebraic conditions (commutation, skew-symmetry, eigenvalue structure) to key navigational properties.

- Interpretability and Transparency
-
The use of matrix exponentials allows analytical examination of conditions such as path invariance, translational invariance, and metric preservation, which are often less accessible in learning-based models.


- Unified Framework for Place, Grid, and Remapping

The same formalism can produce multiple types of spatial tuning depending on parameter choices and eigenvalue structure. The link between roots of unity and grid-like symmetries (e.g., M=2,3) provides a coherent explanation for different spatial patterns.

- Potential for Cross-Domain Generalization

The idea of exponential map representations can, in principle, extend to other domains requiring invariant manifold embeddings (e.g., sensory manifolds, abstract concept spaces).

**Weaknesses:**

- Limited Biological Mechanisms

The framework proposed by this paper is more of descriptive, not prescriptive. It is unclear how actual neural circuits could implement commuting or skew-symmetric generators, or how such matrices could emerge through plausible learning rules.

- Manual Selection of Symmetry Order M

The emergence of different spatial patterns (e.g., square or hexagonal grids) depends on a hand-chosen symmetry order M. Without a mechanism that determines or learns M, the correspondence between algebraic symmetry and biological grid modules remains somewhat ad hoc.

- Relative—but Not Absolute—Control of Grid Scale

The model relates grid spacing ratios to the zeros of the Bessel function, which captures the relative modular scaling between grid modules. However, this construction only determines relative proportions between scales and does not provide a mechanism for setting the absolute grid spacing, which in biological systems depends on physical distance calibration and velocity integration parameters.

**Questions:**

- How might commuting generators arise biologically — e.g., could Hebbian learning enforce approximate commutation under specific motion statistics?

- The model defines modular grid ratios using the zeros of the Bessel function $J_0$. However, $J_0$ is a radial basis in polar coordinates. Does this imply that the resulting representation is egocentric rather than allocentric? If so, how does translational invariance hold under this formulation, and what is the biological interpretation of “origin” in such a coordinate system?

---

> ### Author Response · Authors · 2025-11-25
> **Author Response**
>
> We thank the reviewer for their positive assessment of our theoretical rigor and clarity. We appreciate the insightful questions regarding biological feasibility.
>
> ### 1. Biological implementation and commutativity (Re: Weakness 1 & Question 1)
>
> Regarding biological feasibility, as detailed in the **Global Response** and the new **Appendix H**, we now explicitly map the generator matrices to weights in a Continuous Attractor Network.  We also suggest a local learning rule for learning skew symmetric matrices in **Appendix H**.
>
> ### 2. Grid scale (Re: Weakness 3)
>
> We agree that the relative ratios discussed in the original manuscript do not set the absolute scale. However, in the revised text, we point to a linear relationship between the time constant of this network and the spacing of grid cells generated by the exponential map, in accordance with experimental findings. Also, the Fourier-Bessel series from which we derived the Bessel function zeros features a parameter which may be interpreted as the absolute scale. Specifically, this is the spatial scale at which the similarity decays to zero, and could reflect the scale at which the representation accurately encodes the desired similarity function. We have updated the appendix to highlight this fact.
>
> ### 3. Bessel functions and reference frames (Re: Question 2)
>
> We clarify that the representation is not egocentric in the sense of having a privileged origin. The Bessel function arises in the expansion of the similarity function.  The "origin" in our Bessel function plots refers strictly to zero *displacement* ($\Delta x=0$) in the similarity kernel. Because the similarity depends only on displacement from the current location, a property guaranteed by the skew-symmetry of $G$, the *similarity function* is egocentric. The observed Bessel function structure simply describes the decay of representational similarity as a function of distance arising from the interference of plane waves (roots of unity). This is an isotropic property essential for a metric, independent of the observer's location.
>
> ### 4. Symmetry Order M (Re: Weakness 2)
>
> We acknowledge $M$ is a free parameter algebraically. However, as we discuss in the revised **Limitations** section, external constraints such as maximizing packing density or minimizing path length likely drive biological systems toward $M=3$ (hexagonal). Furthermore, recent work by Pettersen et al. (2024) and Xu et al. (2025) suggests hexagonal grids are optimal for extrinsic distance preservation, providing a functional reason for this selection.

---

### Author Response · Authors · 2025-11-25
**Global Response to Reviewers**

**Note to Reviewers regarding the manuscript PDF:** We have uploaded a revised manuscript. Please note that we are currently finalizing the implementation of your suggestions and the new results discussed below, specifically in the Appendices. We will upload a final, polished version before the discussion period closes.

We thank all reviewers for their constructive feedback. A common theme across reviews was the need to bridge the gap between our algebraic framework and biological mechanisms/learning, as well as to clarify the relationship with prior work such as Successor Representations. In response, we have significantly expanded the manuscript, adding **Appendix H: Biological Interpretation**, **Appendix I: Learning Exponential Maps from Data**, and **Appendix J: Goal-Oriented Navigation**.

### 1. From algebra to biology

Several reviewers asked how the algebraic generators ($G$) could be implemented biologically. Addressing this question, we derive an explicit mapping in the new **Appendix H** between our Exponential Map model and the dynamics of Continuous Attractor Neural Networks (CANNs). We demonstrate that the exponential map describes the on-manifold dynamics of a gain-modulated CANN. Furthermore, by comparing the attractor dynamics $\tau \dot{z} \approx v W z$ with our generator definition $\dot{p} = v G p$, we have included a preliminary result that suggests the network’s time constant scales with hexagonal grid spacing. This provides a normative explanation for the experimental observation that grid spacing expands along the dorsal-ventral axis as stellate cell time constants increase (Giocomo et al., 2007). Finally, we show that a local type update rule can approximate such asymmetric connectivity profiles.

### 2. Connection to other theoretical models of spatial coding

We have updated the text to explicitly discuss McNaughton et al. (2006),  Burak & Fiete, (2009), Stachenfeld et al. (2017), Waniek (2018, 2020), Whittington et al. (2020),  and Rebecca RG et al. (2025). While Successor Representations (SR) models explain grid cells as eigenvectors of a transition matrix (a statistical/predictive view), our framework derives them from algebraic constraints required for path integration (a geometric view). Our condition for translational invariance (skew-symmetric generators, $G^T = -G$) ensures orthogonal transformations, which result in norm-preserving dynamics. This parallels the spectral properties of eigenvectors in translation-invariant transition systems found in SR models. Our framework thus offers a complementary, first-principles derivation that relies on algebraic consistency rather than learned transition statistics.

### 3.  Novelty, predictive power, and the conceptual "punch"

Beyond reproducing grid cell phenomenology, our framework offers a unified, interpretable model of spatial navigation. A key theoretical advance is the demonstration that the exponential map exactly describes the on-manifold dynamics of Continuous Attractor Networks. This yields a concrete prediction: one could fit generator matrices to neural data in a restricted environment and use the learned algebraic structure to accurately predict spatial representations in unseen environments, a capability that remains opaque in standard deep learning models. Furthermore, we demonstrate that grid cell spacing is proportional to the network’s time constant ($\lambda \propto \tau$), matching experimental findings. Finally, by exploiting the remapping operation to orthogonalize representations across contexts, the model supports interpretable, multi-map goal-oriented navigation within a single mathematical framework.

---

### Author Response · Authors · 2025-12-03
**Summary of Contributions and Revisions for the Area Chair**

In light of the recent changes to the review process, we provide this summary to assist the Area Chair in evaluating our submission. This overview highlights the core theoretical contributions of the original manuscript and details how the substantial revisions made during the discussion period have bridged the gap between this theory and biological implementation.

### Core contribution: A first-principles theoretical framework

The primary contribution of this work is the introduction of a transparent, algebraic framework for generating neural spatial representations. Unlike "black-box" deep learning models, where spatial tuning emerges opaquely from optimization, our Exponential Map model constructs representations from a transparent mathematical foundation. By modeling the mapping from physical space to neural activity via the matrix exponential of generator matrices, we identified the exact algebraic conditions that are necessary and sufficient for a coherent neural map of space. The original manuscript established the following key results:

**Exact path integration:** We proved that path independence, the requirement that a location representation depends only on displacement and not the trajectory, is guaranteed if the generator matrices commute.If the generators commute, the model can path integrate exactly and indefinitely. In comparison, existing deep learning models often fail when evaluated on trajectories longer than those observed during training.

**Translational invariance:** We showed that maintaining consistent spatial relationships across locations requires the generators to be skew-symmetric. This ensures orthogonal transformations, resulting in stable, norm-preserving representations ideal for egocentric navigation. Notably, this also enables transparent context-dependent remapping.

**Metric preservation and hexagonal symmetry:** We derived that preserving the flat Euclidean metric imposes a strict spectral constraint: generator eigenvalues must form sets of roots of unity. Crucially, this result provides an algebraic explanation for why hexagonal grid patterns ($M=3$) emerge as optimal solutions for metric-preserving spatial codes.

**Unified mechanism for spatial tuning curves:** We demonstrated that diverse biological phenomena, including periodic grid cells, localized place cells, and context-dependent remapping, can all be constructed from this single mechanism by varying the symmetry of the generators and the structure of the similarity kernel.

### Addressing reviewer concerns with major revisions

While the theoretical foundation was well-received, reviewers (En7c, xaMZ, XtHm, zmDx) requested a stronger connection to biological mechanisms and functional utility. In response, we expanded the manuscript with three new appendices that ground our algebraic findings in neural dynamics and data.

**Biological mechanism (new Appendix H):** Addressing concerns about biological plausibility, we explicitly linked our framework to neuroscience. We now derive the exponential map model as the effective on-manifold dynamics of a gain-modulated Continuous Attractor Neural Network (CANN). Using a Lyapunov energy approach, we show that our abstract generator matrices physically correspond to the velocity-modulated skew-symmetric component of recurrent synaptic weights. This derivation also yields a normative prediction: grid cell spacing must scale linearly with the network's integrative time constant ($\lambda \propto \tau$), matching experimental data on the dorsal-ventral axis (Giocomo et al., 2007).

**Predictive power and extrapolation (new Appendix I):** To demonstrate explanatory power beyond post-hoc fitting, we trained the exponential map model on experimental grid cell recordings (Gardner et al., 2022). The results show that the model not only reproduces training data but successfully extrapolates the spatial code to unseen environments. This confirms that the commutation constraint enables the model to capture the intrinsic algebraic structure of the grid code from noisy biological data.

**Functional navigation (new Appendix J):** We demonstrated that the framework supports interpretable, reward-oriented navigation. By framing the superposition of spatial representations and their orthogonalization via remapping as "bundling" and "binding" operations, analogous to Hyperdimensional Computing, we show that the model supports the storage of multiple reward maps and robust, context-specific retrieval.

We have also updated the manuscript to better contextualize our work within the literature based on recommendations from the reviewers.

### Conclusion
We believe the revised manuscript now offers a comprehensive picture. It combines a solid first-principles theoretical framework, which unifies grid cells, place cells, and remapping under precise algebraic conditions, with a biological derivation that describes the effective dynamics of the neural circuits solving navigation.

---

### Meta-Review · Area_Chair_Tgqo · 2026-01-06

**Summary:**

This paper proposes a new theoretical and computation framework for generating neural spatial representations. The new framework utilizes the exponential map. The resulting models are interpretable and result in many of the hallmark features of spatial navigation such as grid cells.

There were two common concerns raised across the reviews -- is the proposed method biologically plausible and what new insights are gained from it. The authors began to address both of these concerns in the rebuttal, but it is the opinion of the AC that these likely weren't fully addressed.

Specifically, along the axis of biological plausibility, the link between the proposed model and the on-manifold behavior of CANNs is useful and an interesting start, but it doesn't explain everything. The link requires making a constantly velocity assumption (it is unclear if this assumption is always valid or not) and only links to the on-manifold behavior (it is unclear if strictly on-manifold CANNs are biologically plausible or not). Further, it is unclear if the entirety of the proposed exponential map model can be explained by on-manifold CANNs or if on-manifold CANNs are a subset of its expressiveness. Overall, it would be stronger to explain the exponential map model directly as opposed to linking it to one aspect of another computation model.

These are both central parts of the paper but are currently only partially realized and largely relegated to the appendix.

With all this said, I sincerely encourage the authors to continue this work. While the reviewers have valid concerns, there is also genuine excitement about this work and it has the potential to be very impactful and yield novel insights.

**Reviewer Concerns:**

## Reviewer En7c

- Limited applicability to biological mechanisms. I believe this concern is still outstanding.
- Manual selection of symmetry order M. I believe this concern was addressed.
- Control of grid scale. I believe this concern was addressed.

## Reviewer xaMZ

- Clarity of writing. I believe this concern was addressed.
- The contribution of the proposed model compared to existing ones. I believe this concern is still outstanding.
- The behavior observed by changing M is expected. I believe this concern was addressed.
- The comparison to the experimental data is vague. I believe this concern was addressed.

## Reviewer XtHm

- It is unclear what knew knowledge/insights the proposed theoretical model introduces. I believe this concern is still outstanding.

## Reviewer zmDx

- Limited contextulization with prior work. I believe this concern is still outstanding.
- Incomplete biological implementation. I believe this concern is still outstanding.

**Reviewer Scores:**

I do not anticipate that the reviewer scores would have meaningfully changed.

---

### Decision · Program_Chairs · 2026-01-26

Reject